# A theory for self-sustained balanced states in absence of strong external currents

David Angulo-Garcia[1]*, Alessandro Torcini[2,3,4]

1 Universidad Nacional de Colombia - Sede Manizales, Facultad de Ciencias Exactas y Naturales, Departamento de Matemáticas y Estadística, Manizales, Colombia, 2 Laboratoire de Physique Théorique et Modélisation, UMR, CY Cergy Paris Université, CNRS, Cergy-Pontoise, France, 3 CNR - Consiglio Nazionale delle Ricerche - Istituto dei Sistemi Complessi, via Madonna del Piano, Sesto Fiorentino, Italy, 4 INFN Sezione di Firenze, Via Sansone, Sesto Fiorentino, Italy

* dangulog@unal.edu.co

## Abstract

Recurrent neural networks with balanced excitation and inhibition exhibit irregular asynchronous dynamics, which is fundamental for cortical computations. Classical balance mechanisms require strong external input currents in order to sustain finite firing rates, thus raising concerns about their biological plausibility. Here, we investigate an alternative mechanism based on short-term synaptic depression (STD) acting on excitatory-excitatory synapses, which dynamically balances the network activity without the need of strong external driving. Using accurate numerical simulations and theoretical investigations we characterize the dynamics of a densely connected recurrent network made up of $N$ rate-neuron models encompassing STD. Depending on the synaptic strength $J_0$, the network exhibits two distinct regimes: at sufficiently small $J_0$, it converges to a homogeneous fixed point, while for sufficiently large $J_0$ *Rate Chaos* emerges. For finite networks, we observe a *transition region* at intermediate $J_0$, where the system passes from the homogeneous fixed point to Rate Chaos following several different routes to chaos depending on the network realization. Furthermore, we show that the width of the transition region shrinks for increasing $N$ and eventually vanishes in the thermodynamic limit ($N \to \infty$). The characterization of the Rate Chaos regime has been performed by means of Dynamical Mean Field (DMF) approaches. This analysis has revealed on one side that the novel balancing mechanism is able to sustain finite irregular activity even in the thermodynamic limit, and on the other side that balancing occurs via dynamic cancellation of the correlations in the synaptic input currents induced by the dense connectivity. Our findings show that STD provides an intrinsic self-regulating mechanism for balanced networks, sustaining irregular yet stable activity without the need of biologically unrealistic strong external currents. This work extends the balanced network paradigm, offering insights into how cortical circuits could maintain robust dynamics via synaptic adaptation.

**Data availability statement:** All relevant data are within the manuscript and its Supporting information files.

**Funding:** AT received financial support by the Labex MME-DII (Grant No. ANR-11-LBX-0023-01) and by CY Generations (Grant No. ANR-21-EXES-0008) all part of the French program "Investissements d'Avenir". The research has been also partially supported by the ICSC Project: "Centro Nazionale di Ricerca in HPC, Big Data and Quantum Computing - Spoke 8: Insilico Medicine and Omics Data" (CN 00000013– Avviso n. 3138 del 16 dicembre 2021). The funders had no role in study design, data collection and analysis, decision to publish, or preparation of the manuscript.

**Competing interests:** The authors have declared that no competing interests exist.

## Author summary

The human brain is constantly active. This ongoing activity is not random but follows complex patterns that emerge from the interactions between billions of neurons. Understanding how these patterns arise is a fundamental question in neuroscience. One influential idea is that the brain maintains a delicate balance between excitatory and inhibitory signals, preventing runaway activity while allowing rich, flexible dynamics. However, classic theories for this balance mechanism often require strong external inputs to sustain realistic firing rates, which may not agree with biological observations.
Early theoretical work on self-sustained activity in large neuronal networks emphasized the role of intrinsically active neurons as a necessary ingredient to sustain low-rate firing in isolated systems [1]. In contrast, we propose an alternative mechanism based on a biological process called short-term synaptic depression. This process weakens excitatory-excitatory connections when neurons fire too fast, acting as a natural self-regulating mechanism. Using mathematical analysis and computer simulations, we show that this mechanism can maintain stable irregular activity, similar to that observed in the cortex, without the need of strong external inputs. Furthermore, we identify several different paths that our model follows to pass from stable activity to chaotic dynamics, somehow resembling the complex scenarios observed in the brain. Our findings suggest that internal synaptic adaptation may play a key role in shaping neural activity, offering new perspectives on how the brain organizes its complex dynamics.

## Introduction

Neurons in the cortex display highly irregular and asynchronous activity, yet maintaining low average firing rates despite continuous synaptic bombardment. These experimental observations lead initially to the emergence of an apparent paradox: the irregular firing was inconsistent with temporal integration of random excitatory post synaptic potentials (PSPs) [2]. A paradox later solved by introducing the concept of excitation-inhibition balance: neurons in the cortex operate sub-threshold or near threshold due to the balance of the excitatory and inhibitory inputs, therefore the neuronal firing is highly irregular being driven by the fluctuations of the input currents [3,4]. However, it was still unclear how this cancellation of excitation and inhibition can ultimately occur. Balance could eventually occur via a fine tuning of the biological parameters, but this mechanism appears not sufficiently robust in neural circuits, characterized by high heterogeneity. In a seminal work [5], Van Vreeswijk & Sompolinsky proposed in 1996 a dynamical balance mechanism not requiring any fine tuning of the parameters. Balance emerges spontaneously in sparse neuronal networks made of $N$ neurons with mean connectivity (in-degree) $K \ll N$, provided neurons receive a sufficiently large number ($K$) of excitatory and inhibitory pre-synaptic inputs. In such model, asynchronous chaotic activity emerges naturally when the synaptic strengths are sufficiently strong [5,6].

The mechanism underlying dynamical balance relies on the assumptions that synaptic strengths scale as $1/\sqrt{K}$ with the number of presynaptic neurons, and that excitatory and inhibitory PSPs sum linearly to generate the synaptic input currents. This leads to partial excitatory and inhibitory input currents diverging proportionally to $\sqrt{K}$ for increasing in-degree, that should cancel each other to provide finite values for the neuronal firing rates and their fluctuations. The assumption of input linearity requires strong external currents (with amplitudes growing proportional to $\sqrt{K}$) to sustain finite neuronal activity in the limit $1 \ll K \ll N$ [6]. Strong external currents follows from modeling external feed-forward synaptic connections in the same way as the recurrent ones, i.e. by assuming an out-degree $K$ and synaptic strengths $\mathcal{O}(1/\sqrt{K})$ for the pre-synaptic connections due to the neurons in the driving population. If external currents are weak $\mathcal{O}(1)$, excitation and inhibition still balance each other, but the firing activity becomes vanishingly small (as $1/\sqrt{K}$) for increasing connectivity.

The concept of excitation/inhibition balance has been a cornerstone for interpreting neural dynamics in the brain over the last three decades [7]. Moreover, while most of the hypotheses and results of the theory developed in [5] have also been confirmed experimentally [8,9], the assumption that external currents of order $\mathcal{O}(\sqrt{K})$ are necessary to obtain balanced dynamics has recently been challenged [10,11]. These criticisms stem from experimental evidence showing that feed-forward input in cortical circuits is comparable in strength to the total synaptic input [12,13] and therefore of order $\mathcal{O}(1)$ [14–16]. Together with the evidence reported in [9], indicating that sufficiently strong feed-forward stimulation in balanced networks induces a saturation of neuronal responses—possibly due to synaptic depression and/or firing rate adaptation—these findings point toward the need for a novel balance mechanism. Such a mechanism would not require strong external currents but instead rely on some form of synaptic adaptation.

Indeed, a mechanism of this type has been quite recently introduced in [17], where it has been shown for spiking neural networks that the presence of short-term synaptic depression (STD) [18,19] among excitatory neurons suffices to obtain a dynamically balanced regime without the need of any fine tuning and in absence of strong excitatory inputs. The key ingredient allowing for dynamical balance in this case is the fact that STD provides a nonlinear regulatory mechanism able to sustain finite firing rates in an infinitely large network even for external currents $\mathcal{O}(1)$.

In this paper, we investigate in detail the dynamics of firing-rate network models in the presence of this novel balance mechanism. Such balanced networks can exhibit a spectrum of activity regimes, ranging from homogeneous stationary activity to *rate chaos* [20–22]. For finite systems, we observe different transition scenarios from a homogeneous fixed point to rate chaos as the synaptic strength increases. Despite variations across different realizations of the random network, these routes to chaos share a common feature: the initial destabilization of the homogeneous solution, leading to a heterogeneous state (either stationary or oscillatory), followed by a transition region characterized by a complex sequence of dynamical states (e.g., quasi-periodic regimes, stable and chaotic windows). In the thermodynamic limit $N \to \infty$, we conjecture—based on numerical evidence—that the width of this transition region vanishes. Thus, the transition from a stable homogeneous fixed point to rate chaos becomes abrupt at a critical synaptic strength, analogously to what has been reported in [20,21] for rate models and in [22,23] for spiking neural networks with sufficiently slow synaptic dynamics.

Besides detailed numerical investigations, we provide a theoretical description of the dynamical regimes by employing, on one side, Random Matrix Theory and in particular a generalized form of *Girko's circular law* [24–28] to analyze the stability of stationary solutions, and on the other side, Dynamical Mean Field (DMF) approaches [20,29,30] to characterize *rate chaos*. Both approaches are extended to excitatory-inhibitory networks with STD, inspired by previous analyses of excitatory-inhibitory populations [28] and of rate models with frequency adaptation and synaptic filtering [31]. The theoretical results not only reproduce the numerical findings for finite networks with $N \le 10^5$, but also allow the analysis to be extended to extremely large system sizes, up to $N \simeq 10^{12}$, thereby enabling reasonable conjectures about the system's behavior in the thermodynamic limit.

The considered random network is densely connected, i.e., $K \propto N$ [8,32]. For this class of networks, the balancing mechanism in the presence of strong external currents has been reported in [8]. In this context, balance is achieved

through a dynamic cancellation of input correlations induced by the dense connectivity, a phenomenon also confirmed experimentally [9]. In our model with STD, we observe a similar scenario, although external currents are not required to sustain balanced dynamics. Instead, balance is promoted by short-term depression of excitatory-to-excitatory synapses. This form of depression, observed in pyramidal neurons of the visual cortex, has been shown to be the dominant mechanism dynamically regulating the balance between excitation and inhibition to promote stable neural activity [33].

## Results

### Model overview and self-sustained balance mechanism

We consider a recurrent neural network composed of $N$ rate-based neurons divided into two populations: an excitatory population of size $N_E = fN$ and an inhibitory one of size $N_I = (1 - f)N$. The activity of the neurons is governed by the following set of differential equations:

$$\dot{x}_i^E = -x_i^E + \sum_{j \in E}^{N_E} J_{ij}^{EE} \phi[x_j^E] w_j + \sum_{j \in I}^{N_I} J_{ij}^{EI} \phi[x_j^I] + I_0 = -x_i^E + \mu_i^E, \tag{1a}$$

$$\dot{x}_i^I = -x_i^I + \sum_{j \in E}^{N_E} J_{ij}^{IE} \phi[x_j^E] + \sum_{j \in I}^{N_I} J_{ij}^{II} \phi[x_j^I] + I_0 = -x_i^I + \mu_i^I, \tag{1b}$$

$$\dot{w}_i = \frac{1 - w_i}{\tau_D} - u w_i \phi[x_i^E], \tag{1c}$$

where $\mu_i^{E,I}$ denotes the input current to neuron $i$ in the excitatory (E) or inhibitory (I) population, $x_i^{E,I}$ the corresponding exponentially filtered (leaked) input current, $\phi[x]$ the neuronal transfer function that gives the firing rate of the neuron, and $I_0$ a common external excitatory current. Following [19], the effect of STD on the synapses is modeled by the variable $w_i \in [0, 1]$, which dynamically modulates the strength of excitatory-to-excitatory connections. In particular, $w_i$ represents the fraction of resources still available after neurotransmitter depletion, while $u$ denotes the fraction of available resources immediately ready for use. The time scale $\tau_D$ controls the recovery of $w_i$ toward 100% of the available resources.

Each neuron is randomly connected to exactly $K_E = c_E N$ ($K_I = c_I N$) excitatory (inhibitory) presynaptic neurons. Since the connectivity grows proportionally with the system size, the network can be classified as *densely connected* [8,32]. The connectivity matrix **J** has the following block structure:

$$\mathbf{J} = J_0 \begin{pmatrix} \mathbf{J}^{EE} & \mathbf{J}^{EI} \\ \mathbf{J}^{IE} & \mathbf{J}^{II} \end{pmatrix}, \tag{2}$$

where $J_{ij}^{\alpha\beta}$ denotes the synaptic strength exerted by neuron $j$ of population $\beta \in \{E, I\}$ on neuron $i$ of population $\alpha \in \{E, I\}$, and $J_0 > 0$ controls the amplitude of all synaptic strengths.

The non-zero entries of each block are defined as

$$J_{ij}^{EE} = \frac{j_E}{\sqrt{K_E}}, \quad J_{ij}^{IE} = \frac{j_I}{\sqrt{K_E}}, \quad J_{ij}^{EI} = -\frac{g_E j_E}{\sqrt{K_I}}, \quad J_{ij}^{II} = -\frac{g_I j_I}{\sqrt{K_I}}, \tag{3}$$

with $g_E, g_I, j_E, j_I > 0$. The scaling of the synaptic strengths with the in-degrees $K_E$ and $K_I$ in (3) implies that $J_{ij} \propto 1/\sqrt{N}$, as is typically required in balanced networks to ensure that input fluctuations remain $\mathcal{O}(1)$ as $N \to \infty$ [34].

Throughout most of this work we will make use of a transfer function of the following form:

$$\phi[z] = \frac{1}{2}\left(1 + \mathrm{erf}\left(\frac{z}{\sqrt{2}}\right)\right), \tag{4}$$

however, as we will clarify in the following the choice of the transfer function doesn't affect the reported results for sufficiently large system sizes.

Depending on the synaptic strength $J_0$ and the external input current $I_0$, the system exhibits distinct dynamical regimes, which will be thoroughly analyzed in the following sections. Here, we focus on the peculiar dynamical balance mechanism displayed by this network model. Unlike the classical balance mechanism [5,8], which requires strong external input currents to sustain finite firing activity in the large-$N$ limit, the inclusion of nonlinear synaptic dynamics among excitatory neurons enables the emergence of a self-sustained balanced state even when the external input is weak, i.e., $I_0 = \mathcal{O}(1)$ as $N \to \infty$ (as shown in [17] for spiking neural networks).

This new mechanism of dynamical balance can be understood by analyzing the fixed-point (stationary) solutions of the system (1), which are observed at sufficiently small synaptic strengths $J_0$. The fixed point is homogeneous—that is, the equilibrium value is the same for all neurons—and can be determined by solving the stationary equations (1) for a representative neuron in each population. This yields the following reduced set of self-consistent equations:

$$x_0^E = \mu_0^E = \sqrt{N}J_0 j_E \left(\sqrt{c_E}\phi[x_0^E]w_0 - g_E\sqrt{c_I}\phi[x_0^I]\right) + I_0, \tag{5a}$$

$$x_0^I = \mu_0^I = \sqrt{N}J_0 j_I \left(\sqrt{c_E}\phi[x_0^E] - g_I\sqrt{c_I}\phi[x_0^I]\right) + I_0, \tag{5b}$$

$$w_0 = \frac{1}{1 + \tau_D u \phi[x_0^E]}, \tag{5c}$$

where $x_0^\alpha$ is shorthand for the stationary homogeneous solution of population $\alpha \in \{E, I\}$. Since the partial excitatory EE (IE) and inhibitory EI (II) input currents contributing to total excitatory (inhibitory) input current $\mu_0^E$ ($\mu_0^I$) diverge as $\sqrt{N}$, a finite solution can be obtained in the limit $N \to \infty$ only if the terms in parentheses vanish, leading to values of $\mu_0^E$ and $\mu_0^I$ that remain $\mathcal{O}(1)$. This is similar to what happens in the classical theory of balanced dynamics [5,6]. However, in the usual balanced networks, where input currents depend linearly on firing rates, nonzero solutions require strong input currents (i.e., $I_0 \propto \sqrt{N}$). Here, the nonlinear dependence of input currents on firing rates, introduced by STD, allows finite firing rates to emerge even for weak $\mathcal{O}(1)$ or vanishing $I_0$, namely

$$\phi_\infty^E = \frac{1}{\tau_D u}\left(\frac{g_I}{g_E} - 1\right), \tag{6a}$$

$$\phi_\infty^I = \frac{\sqrt{c_E/c_I}}{\tau_D u}\left(\frac{1}{g_E} - \frac{1}{g_I}\right), \tag{6b}$$

$$w_\infty = \frac{g_E}{g_I}, \tag{6c}$$

where $\phi_\infty^\alpha$ denotes the stationary firing rate of population $\alpha$ and $w_\infty$ the stationary synaptic variable as $N \to \infty$. These firing rates remain positive and finite as long as $w_\infty \in [0, 1]$, which is satisfied whenever

$$0 \leq g_E \leq g_I. \tag{7}$$

The balanced steady-state firing rates do not depend on the specific form of the transfer function, but only on the coupling parameters $(g_E, g_I)$, the connectivity densities $(c_E, c_I)$, and the STD parameters. As shown in S1 Appendix, the same thermodynamic-limit solutions are obtained for different classes of transfer functions; the only difference among them lies in the manner in which the large-$N$ limit is approached.

The results reported in (6) refer to the thermodynamic limit, however finite size corrections can be applied to the asymptotic solutions of the firing rates to obtain analytic expressions for the variables that depend explicitly on the parameters $I_0$ and $J_0$, as shown in S2 Appendix. While this new self-sustained balancing regime has here been illustrated for stationary dynamics, it also holds in fluctuating (chaotic) regimes, as we will show in the following.

## Dynamical regimes

The dynamical behaviors displayed by our model as a function of the synaptic coupling strength $J_0$, for fixed values of $I_0$ (which can even be zero, in contrast to spiking networks [17]), are shown in Fig 1 and can be summarized as follows for finite networks. For small $J_0$, the network evolves toward a homogeneous stable fixed point, where all neurons within each population settle to the same steady-state value. As $J_0$ increases, a critical value is reached beyond which the homogeneous fixed point loses stability, giving rise to heterogeneous solutions—either stationary or oscillatory—followed by a transition regime characterized by complex dynamical evolutions. For sufficiently large $J_0$, the system ultimately enters a chaotic regime, commonly referred to as *rate chaos*.

Fig 1 reports the time evolution of neuronal activity for three representative values of $J_0$ in the absence of external drive, i.e., $I_0 = 0$. For small $J_0$ (Fig 1A), all neurons within a population converge to a homogeneous fixed-point value, in excellent agreement with the mean-field predictions (5) (dashed lines). At a critical value of $J_0$ (Fig 1B), the network enters a transition regime where the dynamics become more structured: neuronal activity departs from the homogeneous fixed point and exhibits nontrivial solutions. Depending on the specific finite network realization, this regime may manifest through different intermediate states, characterized by heterogeneous evolutions that can be either stationary or fluctuating. The examples shown in Fig 1B illustrate two typical cases observable at the onset of the transition regime, though many other, more complex dynamical behaviors may arise within this regime, as will be discussed in the following sections. Finally, for sufficiently large $J_0$ (Fig 1C), the system displays strong, irregular fluctuations characteristic of rate chaos.

In the following sections, we provide a detailed analysis of the three dynamical regimes identified above. We first focus on the two main regimes, namely the homogeneous stable fixed-point phase and the chaotic regime, that persist even in the thermodynamic limit. We then characterize the transition region emerging in finite systems. Their properties are investigated using random matrix theory, dynamical system techniques, and dynamical mean-field theory (DMF) [20,29,30]. Finally, we demonstrate that the results obtained for the rate model also hold for a spiking neural network.

## Homogeneous fixed point

We first examine the system size dependence of the homogeneous stationary solution derived in Eq (5). For finite networks the firing rate solutions decrease while the synaptic efficacy $w$ increases with increasing $N$. The theoretical prediction and the network simulations agree almost perfectly up to $N = 100,000$ as seen in Fig 2A. For larger system sizes we rely only on the theoretical predictions (5) to describe the trend of the solutions as they approach the thermodynamic limit. As $N \to \infty$, the solutions converge to the one predicted by the self-balanced argument (6) shown as dashed lines in the same panel. Notice that sizes larger than $N = 10^8$ are needed to achieve the asymptotic results. In our case this amounts to $K_E > 2.5 \times 10^6$ and $K_I > 5 \times 10^5$, definitely larger than the values reported in the cortex, where the estimated value of the (excitatory) synapses is at most of order $10^3$–$10^4$ [10]. We cannot exclude that for other choices of the parameters the convergence to the asymptotic solution could occur for smaller sizes. However, as we will show in the following sub-section Rate Chaos the chaotic solutions converge to their asymptotic balanced values already for $N \simeq 10^6$, which amounts to $K_E \simeq 2.5 \times 10^4$ and $K_I \simeq 5 \simeq 10^3$, i.e. to connectivity values definitely in the range of the ones measured in the cortex.

In Fig 2 B, we report for $N = 10,000$ the variation of the excitatory rate $\phi^E$ as a function of the external current $I_0$ and the synaptic strength $J_0$. On one hand, as expected, increasing the excitatory drive $I_0$ promotes neuronal firing. On the

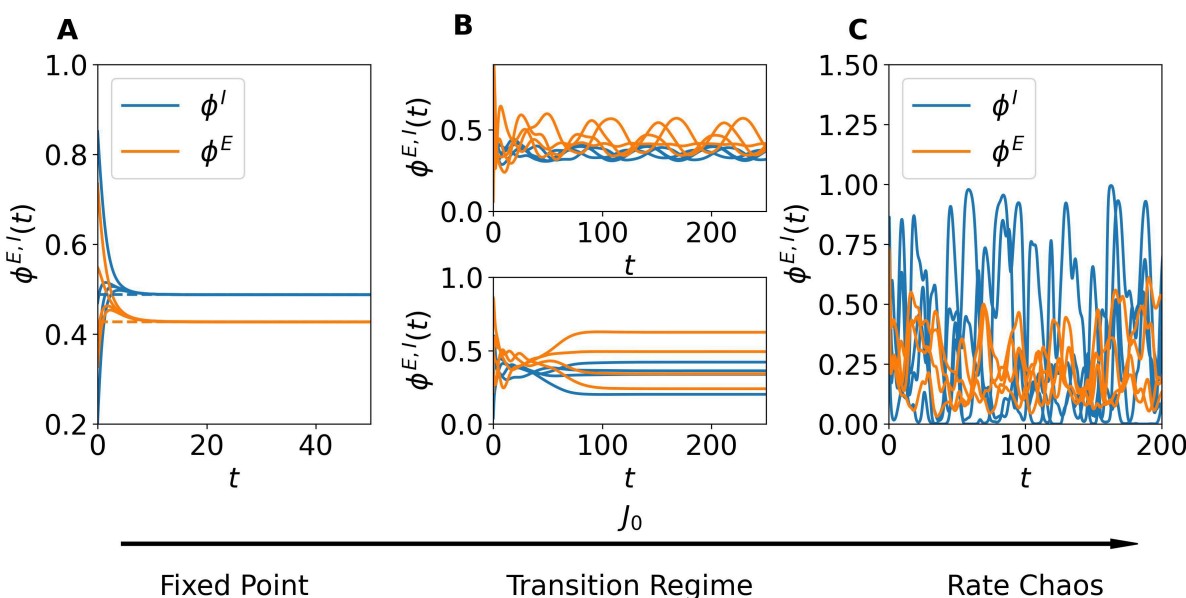

**Fig 1. Dynamical regimes in finite networks.** The panels show the time evolution of the neuronal firing rates for four representative excitatory (orange) and inhibitory (blue) neurons. (A) Homogeneous fixed-point dynamics for low coupling $J_0 = 0.1$. Neuronal activity rapidly converges to homogeneous steady-state values. (B) Transition regime at intermediate coupling $J_0 = 0.87$. Neuronal activity exhibits heterogeneous solutions, either stationary or oscillatory. (C) Chaotic dynamics for strong coupling $J_0 = 1.5$. Neuronal rates display broadband irregular fluctuations. Dashed lines in panel (A) indicate the corresponding mean-field predictions for the population-averaged activity (5). Simulations were performed for network size $N = 10^4$ and no external current ($I_0 = 0$).

other hand, an increase in the synaptic strength $J_0$ leads to a clear reduction of the excitatory firing rate. We have verified that the inhibitory firing rate follows the same trend, with the only difference being that its values are larger than those of the excitatory population, indicating that the network dynamics is inhibition dominated.

The dependence of $\phi^E$ on the parameters $I_0$ and $J_0$ appears inconsistent with the behavior expected in the thermodynamic limit, where firing rates should become independent of both $I_0$ and $J_0$, in accordance with Eqs (6a) and (6b). This apparent discrepancy is resolved by noting that as $N \to \infty$, the influence of the external current and synaptic coupling vanishes. This is illustrated in Fig 2C and 2D, where $\phi^E$ is shown along two cuts of the parameter plane displayed in Fig 2B, for networks of increasing size, namely $10^4 \leq N \leq 10^{12}$. Indeed, for $N = 10^{12}$, the excitatory firing rate stabilizes at its predicted asymptotic value, regardless of the values of $J_0$ or $I_0$, thus confirming the theoretical expectation. The same conclusion holds for $\phi^I$ and $w$. Additionally, the stationary excitatory input current $\mu_0^E$ is reported in the inset of panel C as a function of $I_0$ for the system sizes explored in the main panel. As expected the value of the total input currents are always of the order of $I_0$, despite the partial excitatory and inhibitory input currents diverge proportionally to $\sqrt{K}$ by construction. This is a clear indication that the balance mechanism is at work even at finite $N$, thanks to the action of the depression present in the network.

Furthermore, as discussed in S2 Appendix by adding finite size corrections to the asymptotic balanced solution we obtain expressions for the firing rates that are proportional to $I_0$ and inversely proportional to $J_0$, consistently with the results displayed in Fig 2B–2D.

**Linear stability for homogeneous perturbations.** The stability analysis of the homogeneous fixed point (5) for *homogeneous* perturbations (perturbations equally affecting all the neurons) can be performed by considering the mean

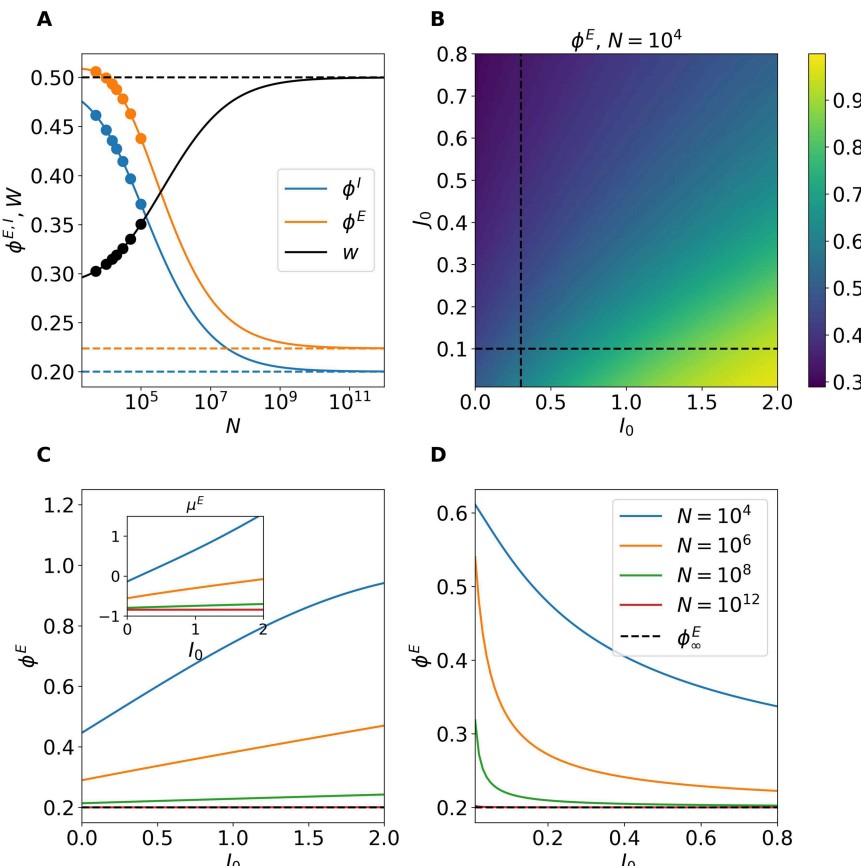

**Fig 2. Finite-size characterization of the homogeneous stationary solutions.** (A) Stationary firing rates and synaptic efficacy as a function of the system size $N$. Symbols correspond to numerical simulations while the solid line shows the self-consistent mean field prediction (5). The solutions obtained in the thermodynamic limit ($\phi_\infty^E$, $\phi_\infty^I$ and $w_\infty$) are reported as dashed lines. Here we set $J_0 = 0.1$ and $I_0 = 0$. (B) Heatmap showing the combined effect of ($I_0, J_0$) on the firing rate of the excitatory neurons in a finite network with $N = 10,000$ obtained by using the mean-field predictions (5). The dashed lines indicate the cuts explored in (C) and (D). (C) Predicted excitatory firing rates at fixed $J_0 = 0.1$ by varying $I_0$ for different network sizes, in the inset are reported the corresponding stationary excitatory input currents $\mu_0^E$. As $N \to \infty$ the effect of the external current becomes negligible. (D) Same as in (C) by fixing $I_0 = 0.3$ and varying $J_0$, these results show the independence of the asymptotic solution from $J_0$.

field-formulation of the network dynamics (1). This reads as

$$\dot{x}_m^E = -x_m^E + J_0 j_E(\sqrt{K_E}\phi[x_m^E]w_m - \sqrt{K_I}g_E\phi[x_m^I]) + I_0 \tag{8a}$$

$$\dot{x}_m^I = -x_m^I + J_0 j_I(\sqrt{K_E}\phi[x_m^E] - \sqrt{K_I}g_I\phi[x_m^I]) + I_0 \tag{8b}$$

$$\dot{w}_m = \frac{1 - w_m^E}{\tau_D} - uw_m\phi[x_m^E]. \tag{8c}$$

In particular, we linearize the above set of equations around the fixed point (5), thus obtaining the corresponding Jacobian matrix -$\mathbf{DF}_{hom}$- and we solve the associated eigenvalue problem (for more details see Linear Stability Analysis for Homogeneous Perturbations in Methods). The real part of the leading eigenvalue Re[$\lambda_{max}$] controls the stability of the homogeneous fixed point for homogeneous perturbations.

In Fig 3A, we plot the real part of the leading eigenvalue as a function of the synaptic strength $J_0$ for different values of the external input $I_0$. For all tested values of $I_0$, Re[$\lambda_{max}$] remains negative, indicating that the homogeneous fixed point is linearly stable to homogeneous perturbations.

To further characterize this trend, we performed a system-size analysis, reported in Fig 3B. In particular, we fixed $I_0 = 2$ and estimated the leading eigenvalues as a function of $J_0$ for $10^4 \leq N \leq 10^{12}$. This analysis shows that the fixed point remains stable for all $0 \leq J_0 \leq 1.1$, and moreover reveals that the real part of the eigenvalue converges to a negative value nearly independent of $J_0$ as $N$ increases. This indicates that, in the thermodynamic limit, the contribution of homogeneous modes to possible instabilities becomes irrelevant. The convergence therefore suggests that homogeneous modes are not involved in the transition to chaos in the thermodynamic limit, which instead originates from the growth of heterogeneous perturbations.

For finite systems, Re[$\lambda_{max}$] could eventually becomes positive for $J_0 > 1.0$ for sufficiently large $I_0$, as suggested by the trend in panel A at $N = 10,000$. However, even if this could occur this instability will not be determinant for the dynamics of the network. Since, as we will show in the next section, for $J_0 > 1$ the system will become unstable due to the growth of heterogeneous modes.

**Linear stability for heterogeneous perturbations.** As we have shown, the homogeneous fixed point is stable under homogeneous perturbations. However, a full stability analysis must also account for heterogeneous perturbations, which affect neurons differently within the same population. To this end, we analyze the eigenvalue problem of the Jacobian **DF**$_{het}$ and employ results from random matrix theory to approximate the eigenspectrum of the system (see Linear Stability Analysis for Heterogeneous Perturbations in Methods).

In summary, extending the results of [28] to the present setting, the spectrum of **DF**$_{het}$ comprises: (i) a bulk of eigenvalues densely distributed within a disk in the complex plane centered at (−1,0) with radius $r$ (a generalization of Girko's circular law [24]); (ii) two outliers, $\lambda_{out}$, which may or may not lie outside that disk; and (iii) a real eigenvalue $\lambda_Q$ with multiplicity $N_E$ whose real part is always negative and therefore does not affect the stability of the fixed point.

For the parameters considered here, the instability arises via the bulk, in line with recent findings in [35]. Whenever $r < 1$, the bulk lies entirely in the left half–plane and the fixed point is linearly stable to heterogeneous perturbations; loss

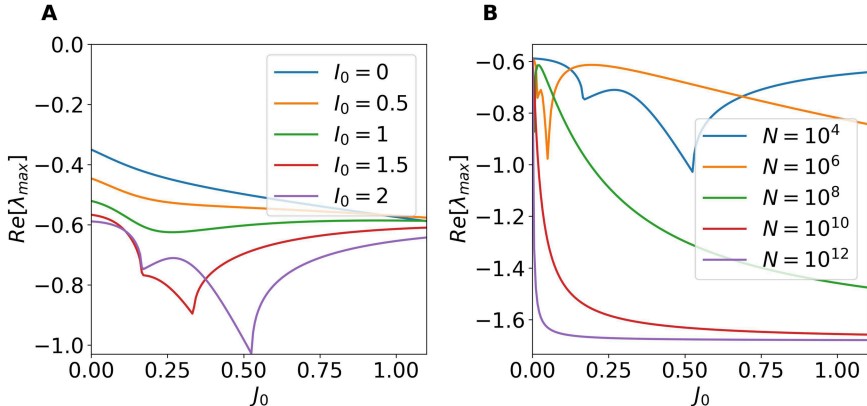

**Fig 3**. **Stability of the homogeneous stationary solutions for homogeneous perturbations.** (A) Real part of the leading eigenvalue of the Jacobian matrix **DF**$_{hom}$ as a function of the synaptic strength $J_0$ for different values of the external DC current $I_0$. For this panel $N = 10,000$ (B) Same as in A as a function of $J_0$ for fixed $I_0 = 2$ and increasing network size $N$.

of stability occurs at $r = 1$. In particular, the radius is given by the following expression:

$$r(J_0) = \frac{J_0}{\sqrt{2}} \sqrt{(a^2 j_E^2 + b^2 g_I^2 j_I^2) + \sqrt{(a^2 j_E^2 + b^2 g_I^2 j_I^2)^2 + 4 b^2 j_E^2 j_I^2 (c^2 g_E^2 - a^2 g_I^2)}},$$ (9)

where $a$, $b$ and $c$ are parameters that depend on the fixed point value (see Linear Stability Analysis for Heterogeneous Perturbations in Methods). In Eq (9) we have expressed the dependence of $r$ only on the the global synaptic strength $J_0$, since we wish to analyze the transitions in terms of $J_0$ by maintaining all the other parameters constant. The condition for the onset of the instability of the homogeneous fixed point is given by the implicit condition $r(J_c) = 1$ that defines the *critical coupling parameter* $J_c$. For further details on the derivation and assumptions involved see the Linear Stability Analysis for Heterogeneous Perturbations in Methods.

In Fig 4A, we compare the eigenvalue spectrum of the full Jacobian $\mathbf{DF}_{\text{het}}$ (computed numerically and shown as blue dots, $\lambda_C$) with the predictions from the random matrix approximation, for $J_0 = 0.1$ and $N = 12,000$. The bulk of eigenvalues is well captured by the circular law, as indicated by the dashed circle labeled "Girko." Additionally, the eigenvalue with multiplicity $N_E$, denoted $\lambda_Q$, is in excellent agreement with the numerical spectrum, as evidenced by the overlap of its marker with the dense cloud of blue points. In contrast, discrepancies are observed for the outlier eigenvalues. However, these discrepancies are not relevant for the stability analysis, since extensive numerical simulations confirm that the outliers do not contribute to any instabilities within the range of parameters considered here (see S3 Appendix in the Supporting Information for a detailed discussion). The onset of instability is instead governed exclusively by the crossing of the imaginary axis of the bulk of eigenvalues enclosed in the disk. Indeed the condition $r(J_c) = 1$ provides an excellent prediction for the critical value $J_c$. This is further confirmed in Fig 4B, where we compare the real part of the largest eigenvalue of the full Jacobian $\mathbf{DF}_{\text{het}}$ with the theoretical prediction obtained by considering the spectral radius $r$ for various values of $I_0$. The agreement between theory and numerical results is excellent across all the tested cases.

An interesting feature observed in Fig 4B is that the critical coupling $J_c$ at which the transition occurs does not vary monotonically with $I_0$. For example, $J_c$ is approximately 0.8 when $I_0 = 0$, decreases for $I_0 = 0.5$, and increases again for $I_0 = 1$ and 1.5. To further investigate this behavior, we computed the critical coupling $J_c$ defined by the condition $r(J_c) = 1$ for increasing network sizes and several values of $I_0$. The results are shown in Fig 4C. As the system size increases, $J_c$ tends to a limiting value, revealing a non-monotonic dependence on $I_0$ for $I_0 > 0$. In all tested cases, $J_c$ converges asymptotically to a value $J_c \approx 1.10$ as $N \to \infty$, independently of $I_0$.

## Rate chaos

We will now focus on the strong coupling regime where rate chaos emerges at sufficiently large values of $J_0 > 1.5$. First, we will apply Dynamic Mean-Field (DMF) approaches [20] to describe the statistical properties of this new balanced state induced by a nonlinear STD term among excitatory neurons. Moreover, we will show that this regime is indeed dynamically balanced either in presence of $I_0 \sim \mathcal{O}(1)$ or even with $I_0 = 0$ and that the mechanisms promoting the balanced dynamics share common aspects with those reported for densely connected recurrent networks with strong external drive [8].

**Dynamic mean field theory.** In the rate chaos regime we can apply Dynamic Mean-Field (DMF) theory to describe the statistical properties of the population-averaged inputs and firing rates. In this framework, the network dynamics is approximated at the mean-field level by single-site excitatory and inhibitory Langevin equations plus an equation for the evolution of the variable controlling the STD on the excitatory single-site neuron, namely:

$$\dot{x}^E = -x^E + \eta^E(t),$$ (10a)
$$\dot{x}^I = -x^I + \eta^I(t),$$ (10b)

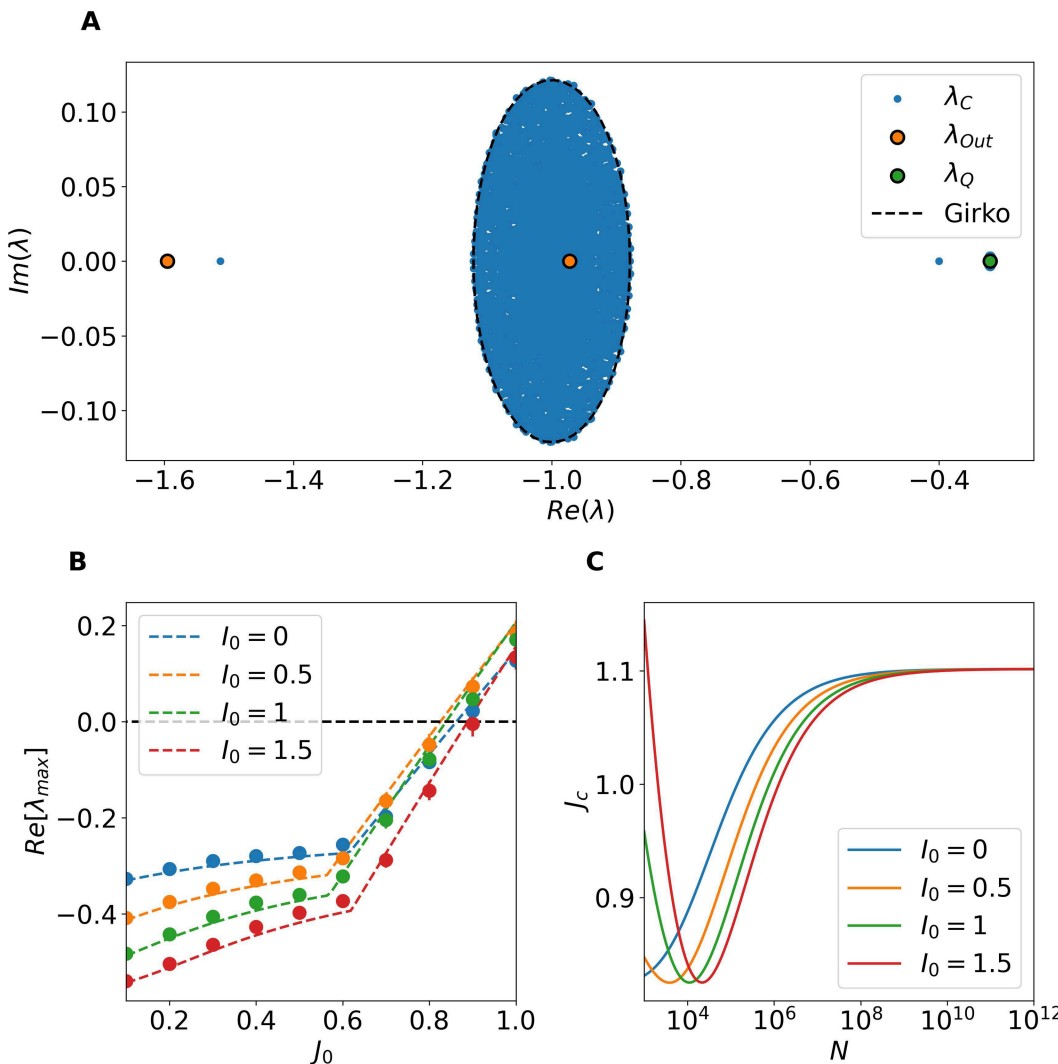

**Fig 4. Stability of the homogeneous stationary solutions for heterogeneous perturbations.** (A) Spectrum of the full Jacobian matrix $\mathbf{DF}_{het}$ (blue dots, $\lambda_C$) compared with predictions from the random matrix approximation. The dashed black circle corresponds to the radius $r$ (25), the orange and green markers indicate the predicted outliers $\lambda_{out}$ and $\lambda_Q$, respectively. In this panel $J_0 = 0.1$ and $N = 5,000$. (B) Maximum real part of the eigenvalue spectrum of $\mathbf{DF}_{het}$ as a function of $J_0$, compared with the radius $r$ predicted by the random matrix approximation for various values of $I_0$. (C) Critical coupling $J_c$ as a function of network size $N$ for different values of $I_0$.

$$\dot{w} = \frac{1-w}{\tau_D} - uw\phi[x^E]; \tag{10c}$$

where $x^\alpha$ now represents the behavior of a generic neuron within the population $\alpha$. The two mean-field neurons experience stochastic Gaussian inputs $\eta^\alpha(t)$, whose statistics are determined self-consistently. In particular the DMF methods yield equations that describe the mean input-currents $\mu^\alpha = [\eta^\alpha]$ as well as the corresponding noise auto-correlation functions (ACFs) for the inputs $[\eta^\alpha(t)\eta^\alpha(t+\tau) - [\eta^\alpha]^2]$ (more details can be found in Dynamic Mean Field Theory in Methods).

In Fig 5A we show, for a network of size $N = 5,000$, representative time traces of the firing rates from four excitatory (orange) and inhibitory (blue) neurons, together with the time traces of synaptic efficacies. As it can be seen, for sufficiently large synaptic coupling ($J_0 = 1.5$), the network enters a strongly fluctuating regime, characteristic of rate chaos.

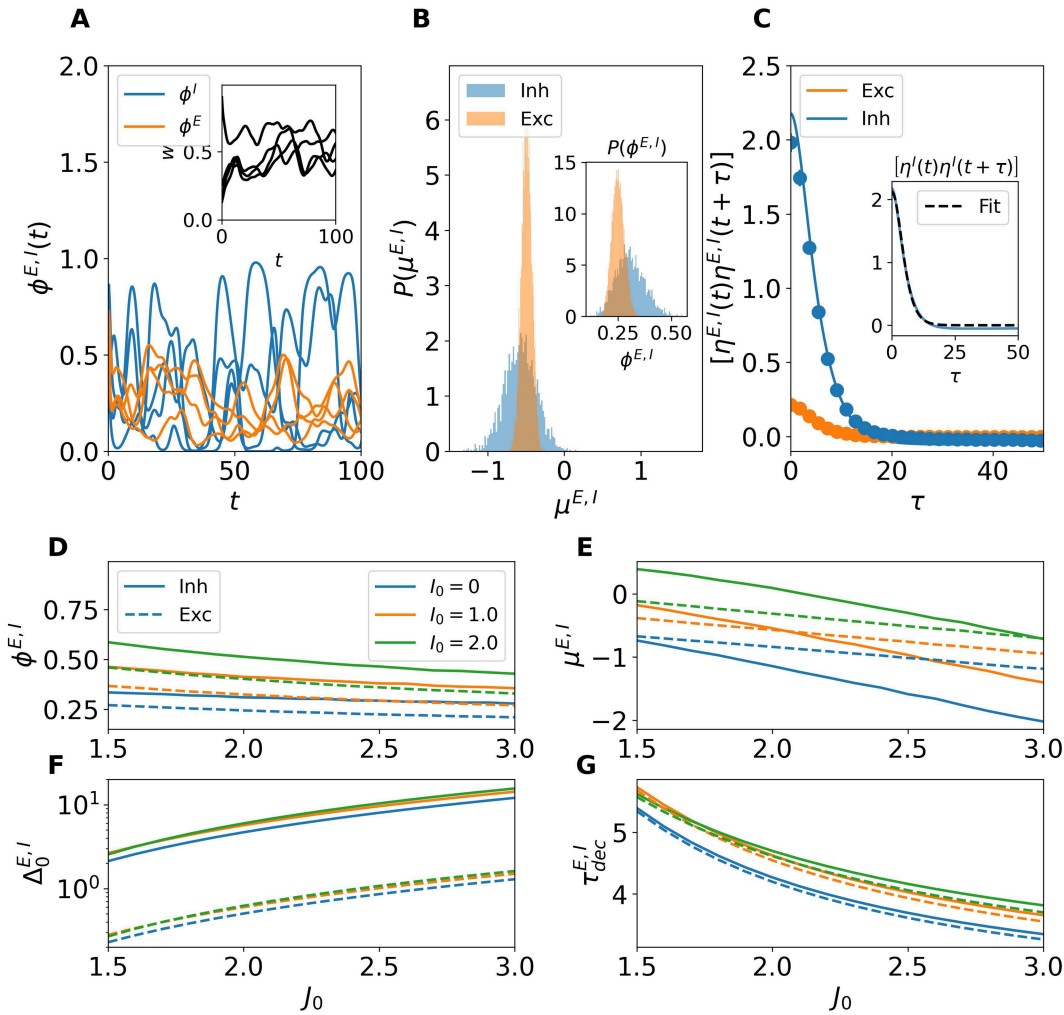

**Fig 5. DMF characterization of the chaotic regime.** (A) Time traces of firing rates $\phi^E(t)$ (orange) and $\phi^I(t)$ (blue) for a subset of neurons in a network of size $N = 5,000$ at $J_0 = 1.5$ and $I_0 = 0$. The inset shows the time traces of the synaptic efficacies for the same excitatory neurons of the main panel. (B) Distribution of total input currents $\mu^{E,I}$ across neurons (main panel) for excitatory (orange) and inhibitory (blue) populations. The inset shows the corresponding firing rate distributions $P(\phi^{E,I})$. (C) Autocorrelation function (ACF) of the total input currents for excitatory and inhibitory populations. Solid lines correspond to DMF predictions and symbols to direct simulations. The inset illustrates a fit of the ACF with the function $\Delta_0 \cosh^2 \tau/\tau_{dec}$ (black dashed lines). (D) Average firing rates, (E) Total input currents, (F) Input variances and (G) Decorrelation times as a function of $J_0$ for different values of the external current $I_0$ (color-coded). In panels (D-G) inhibitory (excitatory) populations are depicted with solid (dashed) lines.

To assess whether DMF approaches are applicable to this regime, we computed the distributions of the time-averaged input currents across neurons (main) and firing rates (inset) for both populations (Fig 5B). The distributions of input currents are well approximated by Gaussian profiles, confirming the applicability of the DMF theory. A clear asymmetry is observed between populations: inhibitory inputs exhibit much larger variability than excitatory ones. This difference translates into distinct firing rate distributions: while both populations exhibit a bell shaped distribution of the firing rates, inhibitory populations present a broader distribution of the firing rates with a larger mean value.

Having verified that the Gaussian assumption holds, we used DMF theory to compute the autocorrelation function (ACF) of the total input currents for both populations. These are reported in Fig 5C, where solid lines represent DMF predictions and symbols numerical simulations, showing excellent agreement. The inhibitory ACF displays a higher peak

amplitude compared to the excitatory ACF, consistent with the stronger fluctuations observed in panels A and B. A small mismatch at $\tau = 0$ reflects the finite-size nature of the simulations, since DMF is exact only in the thermodynamic limit ($N \rightarrow \infty$; for more details see S4 Appendix).

We then systematically investigated the effect of varying $J_0$ and $I_0$ on four key indicators: mean firing rates $\phi^{E,I}$ (Fig 5D), total input currents $\mu^{E,I}$ (Fig 5E), variances of the total input currents $\Delta_0^{E,I}$ (Fig 5F), and decorrelation times $\tau_{\text{dec}}^{E,I}$ (Fig 5G). As shown in Fig 5D, the firing rates systematically decrease with stronger synaptic coupling $J_0$, thus confirming that our balanced network is operating in a inhibition dominated regime. However, increasing values of the external drive $I_0$ counteracts this decrease in both populations. In Fig 5E, the total input currents also decrease (increase) with $J_0$ ($I_0$) in agreement with what observed for the firing rates.

The variances $\Delta_0^{E,I}$, reported in Fig 5F, grow markedly with $J_0$ and are only weakly affected by $I_0$, suggesting that $I_0$ is not particularly involved in the balance mechanism. It should be noticed that fluctuations in the inhibitory input currents are consistently almost an order of magnitude larger than excitatory ones. Finally, Fig 5G shows that the decorrelation time $\tau_{\text{dec}}$ decreases with stronger coupling $J_0$, indicating that the fluctuation dynamics becomes faster. The external input current $I_0$ tends to slightly increase $\tau_{\text{dec}}$, though in the narrow interval $1.5 < J_0 < 1.7$ a non-monotonic behaviour emerges: the longest decorrelation time is reached for $I_0 = 1.0$, and decreases slightly for $I_0 = 2.0$. Beyond $J_0 > 1.7$, the dependence becomes monotonic with decorrelation times always increasing for growing $I_0$. Notably, $\tau_{\text{dec}}$ is very similar for excitatory and inhibitory populations, although differences gradually widen with increasing $J_0$, suggesting a subtle asymmetry in their temporal response to stronger coupling.

Altogether, these results demonstrate that DMF theory provides a consistent and accurate description of chaotic dynamics in balanced networks even for finite system sizes, where structured yet irregular activity emerges with fluctuation amplitudes and correlation times strongly influenced by the synaptic coupling and external drive. However, as we will show in the following the influence of the external current will vanish for sufficiently large $N$.

Analogously to the analysis performed for the *Homogeneous Fixed Point*, we now use the DMF approach to verify whether the proposed balancing mechanism can sustain finite firing activity also in the rate chaos regime, even in the absence of strong external inputs. Specifically, we study the system's behavior as the network size $N$ increases up to $10^{10}$, while fixing $J_0 = 1.5$ and $I_0 = 0$. The results are reported in Fig 6.

Fig 6A displays the population-averaged firing rates $[\phi^{E,I}]$ and synaptic efficacy $[w]$. As in the fixed point case, the agreement between direct simulations (symbols) and DMF predictions (solid lines) is excellent up to $N = 128,000$. Beyond this point, we rely solely on DMF predictions. As $N$ increases further, the variables approach asymptotic values, stabilizing around $N = 10^8$ (dashed lines), despite the absence of any external input.

As shown in Fig 6B, the DMF results for the mean inputs $\mu^{E,I}$ converge to definitely negative values in the large $N$ limit both for excitatory and inhibitory neurons with $\mu^I < \mu^E$ corresponding to finite firing rates as shown in panel A. On the other hand, as shown in the inset of panel B, also the variances $\Delta_0^{E,I}$ converge to finite values. Furthermore, the fact that $\Delta_0^I > \Delta_0^E$ indicates that fluctuations are larger in the inhibitory populations with respect to excitatory ones as reported in the previous figure for finite size networks.

Additionally, Fig 6C shows the autocorrelation function of the total input to the excitatory population for various network sizes. Smaller networks exhibit larger autocorrelation peaks (i.e., larger variances) in agreement with the results in the inset of panel B. However, for sufficiently large system sizes ($N > 10^8$) the autocorrelation functions converge to an asymptotic profile. On the other hand, the temporal decay remains largely unchanged, with a decorrelation time $\tau_{\text{dec}}^{E,I} \approx 6$ for all system sizes.

The mechanism by which the system generates finite synaptic inputs despite $N \rightarrow \infty$ can be understood by using arguments analogous to those used in the fixed point analysis. In the chaotic regime, the mean input currents can be expressed as

$$\langle \mu^E \rangle = \sqrt{N} J_0 j_E (\sqrt{c_E} \langle \phi[x^E] w \rangle - g_E \sqrt{c_I} \langle \phi[x^I] \rangle) + I_0 = \sqrt{N} J_0 j_E A^E + I_0, \tag{11a}$$

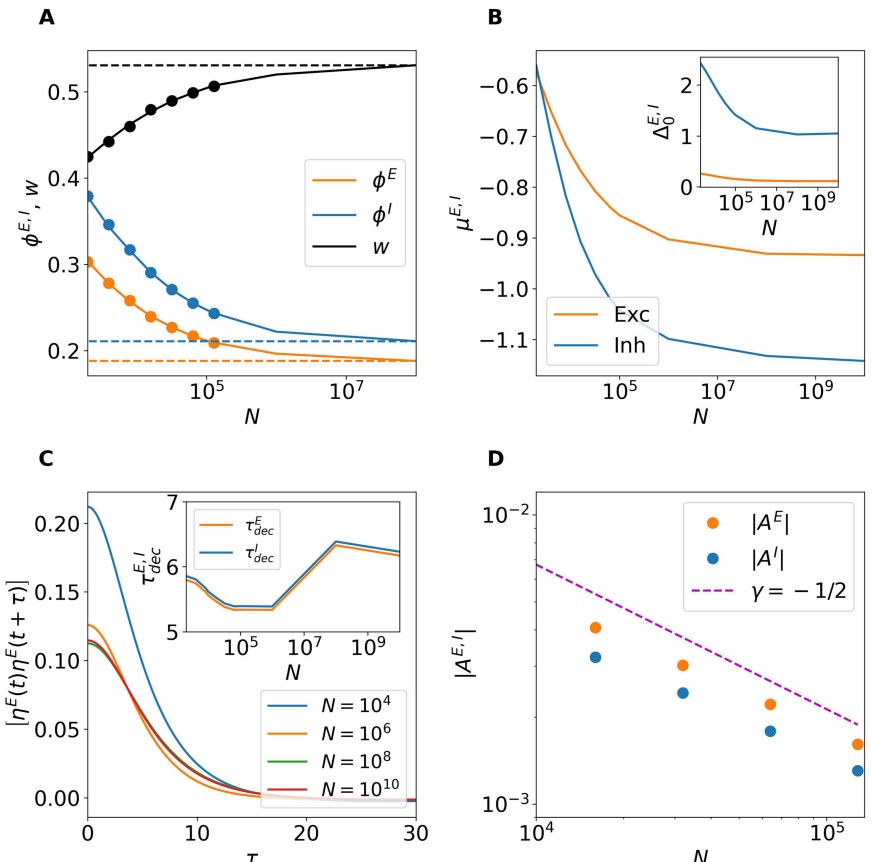

**Fig 6. Rate chaos approaching the thermodynamic limit.** (A) Average firing rates [$\phi^{E,I}$] and synaptic efficacy [$w$] as a function of network size $N$. Symbols refer to direct network simulations, while solid lines represent DMF predictions. (B) Variances $\Delta_0^{E,I}$ of the total inputs for increasing $N$ predicted by DMF theory. (C) Autocorrelation function of the excitatory synaptic input for different $N$. Inset: Decorrelation times $\tau_{\text{dec}}^{E,I}$ versus $N$. (D) $|A_E|, |A^I|$ versus $N$. The magenta dashed line indicates a power law decay $N^{-1/2}$. For this figure all simulations were performed by averaging a time interval $t = 1{,}000$, with $I_0 = 0$ and $J_0 = 1.5$.

$$\langle \mu^I \rangle \;\; = \;\; \sqrt{N}J_0 j_I(\sqrt{c_E}\langle\phi[x^E]\rangle - g_I\sqrt{c_I}\langle\phi[x^I]\rangle) + I_0 = \sqrt{N}J_0 j_I A^I + I_0, \tag{11b}$$

where $\langle\cdot\rangle$ denotes averaging over the corresponding neuronal population, time, and potentially across network realizations. For the balancing mechanism to be effective, the terms inside the parentheses—denoted $A^E$ and $A^I$—must vanish as $1/\sqrt{N}$ in the large-$N$ limit, thereby compensating for the divergence of the $\sqrt{N}$ prefactor [6]. Fig 6D confirms that this scaling of $|A^{E,I}|$ with $N$ is indeed realized, demonstrating that the balancing mechanism works even in the absence of an external input ($I_0 = 0$ in this case) thanks to the synaptic depression acting on the excitatory neurons.

**Balancing mechanism in densely connected recurrent networks.** Let us now perform a more refined analysis of the functioning of the balancing mechanism, as we stated initially in our model the in-degrees $K_E$ and $K_I$ grow proportionally to $N$, therefore our model is not sparsely connected, as the model examined in [5], but it is a random densely connected network accordingly to the definition given in [32]. The emergence of dynamical balance in densely connected networks have been examined in [8]. In such a paper, the authors observed that the correlations among partial input currents (either excitatory or inhibitory) stimulating different neurons remains finite in the large $N$ limit, as expected

due to the sharing of common inputs. However, the total input currents are weakly correlated as well as the firing activity of the neurons. This has been explained as due to a dynamic cancellation of input currents correlations obtained via a tracking of the fluctuations in the excitatory (inhibitory) partial input currents (for definitions and more details see Correlation Coefficients of the Input Currents and of the Firing Rates in Methods).

As shown in Fig 7A, also in the present case the partial input currents are definitely correlated, furthermore the corresponding correlations show a trend to grow with the system size. For sufficiently large system sizes one eventually would expect all the correlations to saturate, however this limit size is not yet reached. In particular, one observes that the correlation coefficients related to excitatory neurons $\rho^{EE}$ and $\rho^{IE}$ are definitely larger than the ones related to inhibitory neurons $\rho^{EI}$ and $\rho^{II}$. This difference will be partially compensated at larger $N$, since $\rho^{EE}$ and $\rho^{IE}$ grows as $N^{\gamma}$ with an exponent $\gamma \sim 0.15$, while $\rho^{EI}$ and $\rho^{II}$ grows much faster with an exponent that is the double, namely $\gamma \sim 0.30$. The higher level of correlation among the excitatory partial input currents cannot be simply explained by the fact that the connectivity is higher among excitatory neurons (3.1%) with respect to inhibitory ones (2.5%). This difference is probably due to the fact that $N_I = N_E/4$ and larger system sizes are required to achieve similar level of correlations among inhibitory input currents.

As expected for sufficiently large $N$ the correlations $\rho^{EI}$ and $\rho^{II}$ coincide, this is not the case for the correlations of the excitatory partial input currents. Indeed, the effect of the STD, present in the excitatory currents stimulating excitatory neurons $h_i^{EE}$, induces a higher level of correlation among these terms with respect to the excitatory currents stimulating inhibitory neurons $h_i^{IE}$, where STD is absent. As a matter of fact, the values of $\rho^{EE}$ are roughly the double of those of $\rho^{IE}$ for all the examined system sizes, namely $2,000 \leq N \leq 80,000$.

A striking difference can be observed by considering the correlation coefficients of the total inputs currents $\rho_T^E$ and $\rho_T^I$ since these are clearly smaller and decrease with $N$, as shown in the main panel of Fig 7B. In particular, these Pearson coefficients decay as $N^{-\gamma}$ with $\gamma \simeq 0.70 - 0.75$ for $N \geq 20,000$. This is a clear effect of the dynamic cancellation of the input currents correlations achieved via a nonlinear balancing mechanism. For the usual balancing mechanism the authors in [8] reported a theoretical argument for which the correlations of the total input current should vanish as $N^{-1/2}$, whenever $(\rho^{EE}, \rho^{EI}, \rho^{IE}, \rho^{II})$ saturate. In the present case, despite the Pearson coefficients $(\rho^{EE}, \rho^{EI}, \rho^{IE}, \rho^{II})$ are still slightly growing

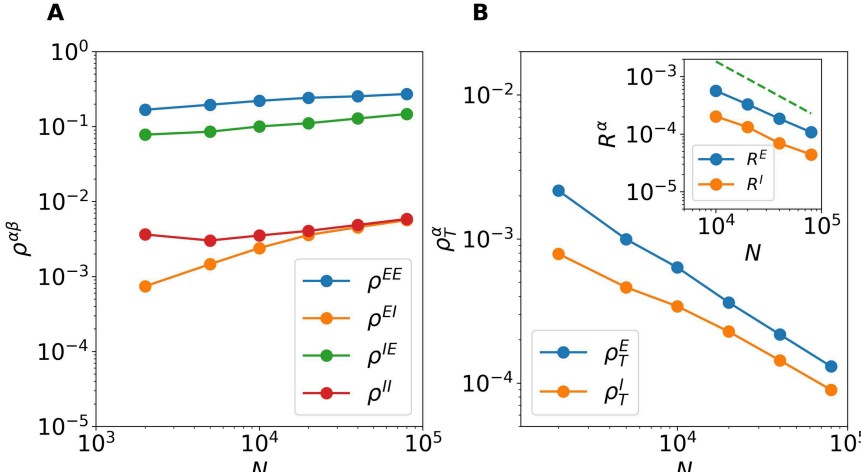

**Fig 7. Correlations in the balanced regime.** (A) Population averaged Pearson correlation coefficients ($\rho^{EE}, \rho^{EI}, \rho^{IE}, \rho^{II}$) among the partial input currents versus $N$. (B) Population averaged Pearson coefficients ($\rho_T^E, \rho_T^I$) of the total input currents. Inset: Correlation coefficients $R^E$ and $R^I$ of the excitatory and inhibitory firing rates versus the system size $N$. The red dashed line refer to a power law decay $1/N$. The data are averaged over 10 different realizations of the random network, and over a time interval $t = 300$, after discarding a transient of duration 500. For this figure $I_0 = 0$ and $J_0 = 1.5$.

with $N$, $(\rho_T^E, \rho_T^I)$ decay faster than $N^{-1/2}$ suggesting that the present balance mechanism is more effective than the classical one.

As a final aspect, we consider correlations among the firing rates, for an asynchronous regime we expect that the population averaged correlation coefficients of the firing rates $R^E$ and $R^I$ will vanish for sufficiently large system sizes as $1/N$ (due to the central limit theorem). Indeed $R^E$ and $R^I$ will vanish for $N \to \infty$, as shown in the inset of Fig 7B. The power law decay is not $1/N$ as expected, but it can be reasonably well fitted with a power law $N^{-0.8}$ within the examined ranges of system sizes. The agreement with the expected behavior (red dashed line in panel C) can be considered as consistent, larger system sizes would be required to obtain a better agreement, however this goes beyond our computational capabilities.

### Transition scenarios towards rate chaos in finite systems

**Bifurcation mechanisms at the onset of the instability.** Having characterized the two main dynamical regimes of the model—the homogeneous stable fixed point at low coupling and the rate chaos regime at strong coupling—we now turn to the intermediate region separating them. This transition region, which is observed only in finite networks, plays a crucial role in shaping the dynamical pathways toward chaos and provides insight into how the thermodynamic-limit behavior is approached as the system size increases.

For finite systems, we have observed two distinct bifurcations via which the homogeneous stationary solution can lose stability. Both are characterized by the emergence of heterogeneous solutions, that can be either oscillatory or stationary. The observed bifurcation mechanism depends on the realization of the random network.

The first transition pathway corresponds to a classical Hopf bifurcation (see Fig 8A). In this case, a pair of complex conjugate eigenvalues crosses the imaginary axis, giving rise to stable oscillatory dynamics. This is characterized by the emergence of heterogeneous periodic modulations of the firing rate.

The second pathway is mediated by a zero-frequency bifurcation, in which a single real eigenvalue crosses zero. This scenario destabilizes the homogeneous fixed point and gives rise to a heterogeneous fixed point (see Fig 8B). The new attractor is characterized by Gaussian-distributed input currents $\{\mu_i^E\}$ and $\{\mu_i^I\}$, which in turn generate neuron-specific firing rates that break the population symmetry. Interestingly, the parameters of the stationary distributions of the input currents associated with the heterogeneous fixed point can be derived self-consistently by solving for $\mu^{E,I}$ together with their corresponding variances $\Delta_0^{E,I}$ (see Statistics of the Heterogeneous Fixed Point in Methods). In particular, Fig 8C shows the distribution of synaptic input currents for the excitatory population, obtained from a network realization that converges to a heterogeneous fixed point, together with the corresponding Gaussian profile predicted by the self-consistent set of equations (53), (54), (55), and (56) reported in Statistics of the Heterogeneous Fixed Point within Methods. The inset displays the distributions of synaptic efficacies obtained from direct simulations and compares them with the theoretical prediction given in (58) in Statistics of the Heterogeneous Fixed Point. The agreement between network simulations and theoretical predictions is remarkably good for both quantities.

Finally, Fig 8D shows the self-consistent solutions for $\mu^E$ (main panel) and $\Delta_0^E$ (inset) as functions of $J_0$. For $J_0 < J_c$, the self-consistent framework correctly reproduces the homogeneous solution (orange line), characterized by vanishing variances (see inset). For $J_0 \geq J_c$, the solution bifurcates: the unstable homogeneous branch corresponds to less negative values of the mean excitatory input current, whereas the stable heterogeneous fixed point is associated with more negative $\mu^E$ and an increasing $\Delta_0^E$ as the synaptic coupling grows.

The heterogeneous fixed point and the oscillatory solutions discussed here remain stable only within a narrow range of synaptic coupling strengths, before undergoing further instabilities that will be analyzed in the following section.

**Routes to rate chaos.** Following the initial destabilization of the homogeneous fixed point — either via a Hopf bifurcation or a zero-frequency bifurcation, as described in the previous subsection — the system may follow different dynamical

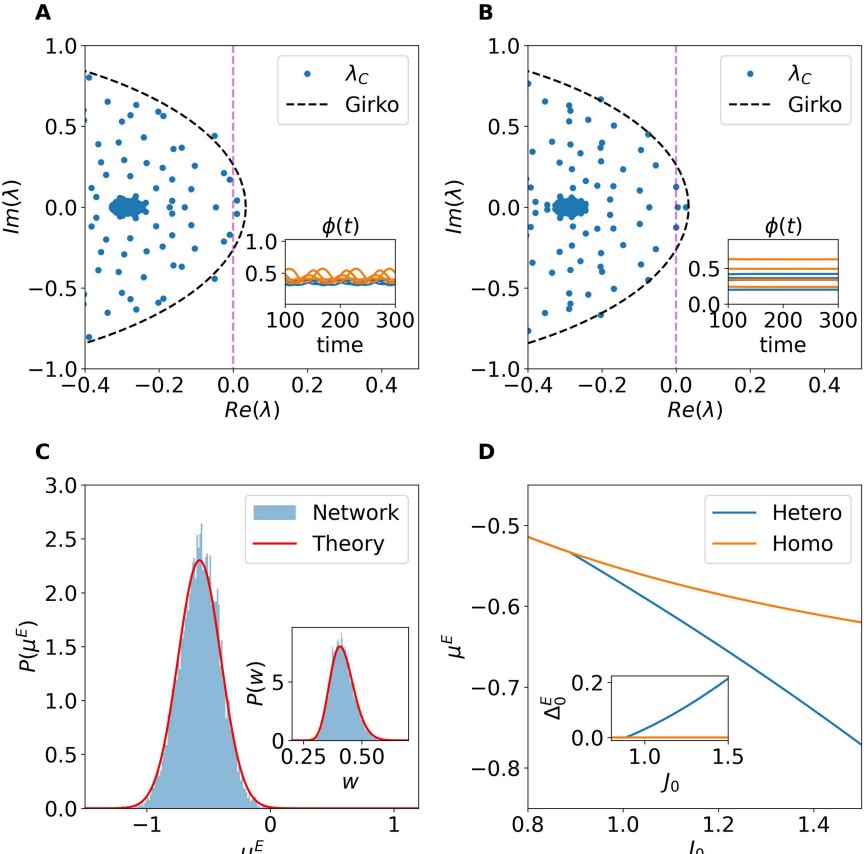

**Fig 8. Two distinct bifurcation mechanisms driving the instability of the homogeneous fixed point.** (A) Hopf bifurcation: two complex conjugate eigenvalues crosses the imaginary axis. (B) Zero-frequency bifurcation: One real eigenvalue crosses zero, leading to a stationary heterogeneous solution. In the insets in (A-B) are reported the firing activity of few excitatory (inhibitory) neurons above the corresponding transition displayed in orange (blue). (C) Distribution of the input currents for the excitatory population in a network simulation with heterogeneous fixed point (blue shaded histogram) and the Gaussian theoretical prediction (red line). Inset: Corresponding distribution of the synaptic efficacies. For panels A-C we have used $J_0 = 1.0$ and $I_0 = 0$. (D) Theoretical prediction for the average input current (main) and the standard deviation (inset) as a function of the synaptic coupling $J_0$. These correspond to the self-consistent solutions of Eqs (53) and (54). For all the panels in the figure we have considered $N = 10,000$.

pathways before reaching the fully chaotic regime. These routes to chaos are strongly influenced by the specific realization of the random network connectivity and can display a variety of intermediate states.

To systematically investigate these routes, we have computed the two largest Lyapunov exponents (LEs) $\Lambda_1$ and $\Lambda_2$ as a function of the synaptic strength $J_0$, see Lyapunov Analysis in Methods for their definition. The LEs provide a quantitative measure of the system's sensitivity to initial perturbations and allows to classify the possible dynamical regimes. In particular, a fixed point is associated to $\Lambda_2 < \Lambda_1 < 0$; a periodic regime to $\Lambda_1 = 0$ and $\Lambda_2 < 0$; a quasi-periodic dynamics on a Torus $T^2$ to $\Lambda_1 = \Lambda_2 = 0$ and chaos to at least $\Lambda_1 > 0$ [36].

Four representative examples of these routes to chaos are reported in Fig 9A–9D. In all cases, the system initially resides in a stable fixed point ($\Lambda_1 < 0$). At the critical coupling $J_c$, it enters a transition region which, depending on the network realization, may begin either with heterogeneous stable oscillations or with a heterogeneous fixed point, as discussed in the previous subsection. From there, the dynamics evolve through a variety of increasingly complex regimes before reaching chaos at larger $J_0$.

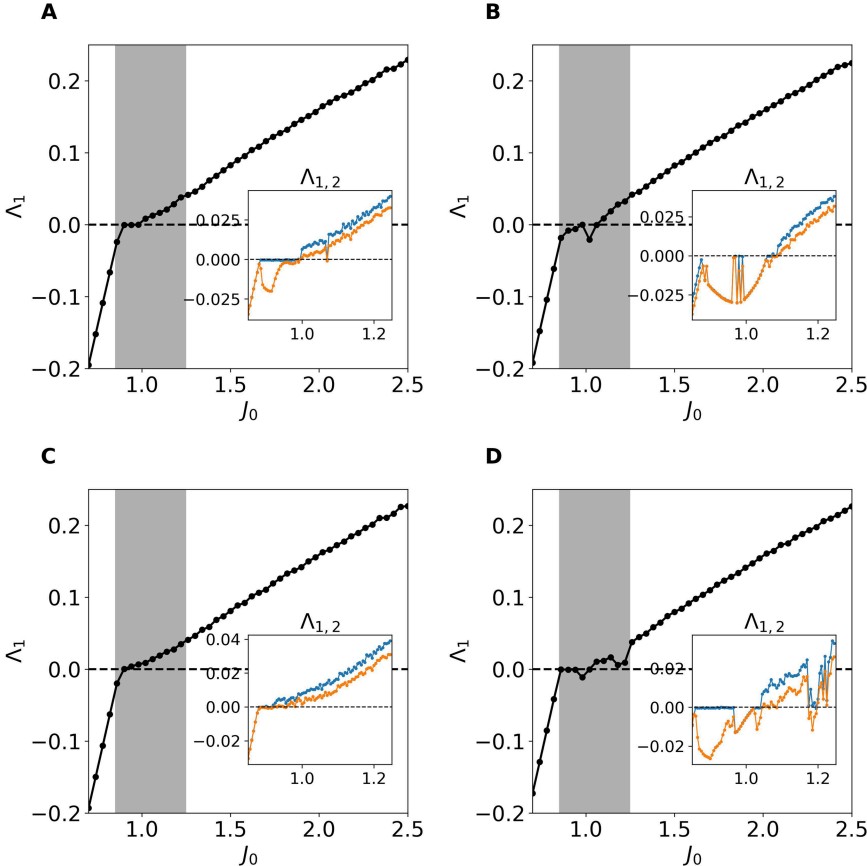

**Fig 9. Lyapunov characterization of the routes to chaos.** (A–D) Main Figures: Maximal LE $\Lambda_1$ as function of $J_0$ with $I_0 = 0$. After the fixed point loses stability at $J_0 = J_c$ a transition regime emerges -shaded area- where, depending on the network realization different routes to chaos can be identified. In the inset of each figure we show the first and second largest LEs calculated with a higher resolution in $J_0$ with the aim of characterizing the transition region towards chaos.

In some cases (see Fig 9A), the transition proceeds through a Hopf bifurcation, leading to stable oscillations ($\Lambda_1 = 0$, $\Lambda_2 < 0$), which eventually give rise to high-dimensional chaos ($\Lambda_1, \Lambda_2 > 0$). In other cases (see Fig 9B for an example), the system passes from a homogeneous to a heterogeneous fixed point ($\Lambda_1 < 0$, $\Lambda_2 < 0$) before becoming chaotic.

Additional scenarios include quasi-periodic dynamics (Fig 9C) ($\Lambda_1 = 0$, $\Lambda_2 = 0$) and more intricate routes with interplay of chaotic ($\Lambda_1 > 0$) and stability windows (Fig 9D).

These results show the richness of the transition regime and the relevant role played by finite-size effects. While the final outcome is always rate chaos at sufficiently large $J_0$, the pathway leading there is not unique. A complete classification of all possible transitions is beyond the scope of this work, but these findings provide insights into the diverse routes to chaos that can emerge in finite size networks for the studied model.

Moreover, our numerical analysis shows that the width of the transition region systematically decreases with system size (see S5 Appendix in Supporting Information), thus suggesting that in the thermodynamic limit the transition from a stable homogeneous fixed point to chaos becomes abrupt. This behavior is consistent with previous reports for *classically* balanced rate models [21] as well as for spiking neural networks with sufficiently slow synaptic dynamics [22,23].

## Spiking neural network

Finally, we test whether the results presented so far also hold for a more realistic spiking neural network. To this end, we simulate a Leaky Integrate-and-Fire (LIF) network coupled via exponentially decaying post-synaptic potentials, with the assumption that the synaptic decay time is much larger than the membrane time constant, i.e., $\tau_{syn} \gg \tau_m$ (see Spiking Neural Network: Leaky Integrate-and-Fire Neurons with Synaptic Dynamics in Methods for more details on the network model). Under this assumption, the dynamics of the network can be approximated by the following rate model [21,22]:

$$\tau_{syn}\dot{x}_i^E = -x_i^E + \sum_{j \in E}^{N_E} J_{ij}^{EE}\phi[x_j^E]w_j - \sum_{j \in I}^{N_I} J_{ij}^{EI}\phi[x_j^I] + I_0 \tag{12a}$$

$$\tau_{syn}\dot{x}_i^I = -x_i^I + \sum_{j \in E}^{N_E} J_{ij}^{IE}\phi[x_j^E] - \sum_{j \in I}^{N_I} J_{ij}^{II}\phi[x_j^I] + I_0 \tag{12b}$$

$$\dot{w}_i = \frac{1 - w_i}{\tau_D} - uw_i\phi[x_i^E] \tag{12c}$$

where the transfer function is given by

$$\phi[z] = -\frac{1}{\tau_m \log(1 - 1/z)}. \tag{13}$$

It is important to note that for the LIF model a neuron is sub-threshold (supra-threshold) whenever $z < 1$ ($z > 1$).

The assumption $\tau_{syn} \gg \tau_m$ ensures that the *noise* due to the spiking activity is effectively filtered out on a time scale $\tau_{syn}$, allowing for the theoretical framework developed for the rate model to be applied. In particular, for small synaptic coupling we observe a stable homogeneous fixed point for the rate model (12) with constant supra-threshold input currents $\mu^E > 1$ and $\mu^I > 1$, this would correspond to neurons firing with equal mean firing rates in the network model (62) (63). For sufficiently large synaptic coupling a rate chaos regime emerges in both the spiking LIF network as well as in the corresponding rate model.

Fig 10 summarizes the dynamics in the fixed point regime. In particular Fig 10A shows a raster plot from the spiking network simulation, displaying tonic firing for each neuron, consistently with the supra-threshold inputs reported in Fig 10B for an excitatory and an inhibitory sample neuron. In panel B, we also compare these input currents obtained from the spiking network to the mean-field predictions for excitatory and inhibitory neurons (dashed lines), observing a reasonable agreement. Notice that the actual input for the neuron in the spiking network is quite noisy due to the spiking mechanism. However, the mean input currents for each neuron are quite similar and the corresponding mean firing rates present a very small variability with a variance across the neurons $\approx 1 \times 10^{-3}$ clearly corresponding to an homogeneous fixed point solution in the rate model. Fig 10C shows the largest eigenvalue of the heterogeneous perturbation Jacobian $\mathbf{DF}_{het}$ (symbols) and the prediction from random matrix theory (solid lines) for various values of external input $I_0$ using the rate model with LIF transfer function. The results show a close match between the theory and the numerical diagonalization of the Jacobian matrix, accurately predicting the critical coupling strength $J_c$ for the destabilization of the homogeneous fixed point. We also observe from panel C that for finite $N$ (namely, $N = 5,000$) increasing $I_0$ shifts the critical coupling strength to higher values. However, by using the generalized Girko's circular law, we observe that for sufficiently large $N > 10^9$ the critical coupling strength converges to a value largely independent of $I_0$ (namely $J_c \approx 0.075$) as expected, (see Fig 10D).

Next, we analyze the dynamics of the spiking network in the chaotic regime for sufficiently large coupling strength, namely $J_0 = 0.4$. As shown in Fig 11, the network exhibits irregular bursting activity interspersed with quiescent periods (Fig 11A), a hallmark of the dynamics in networks with dominant inhibition and long synaptic timescales [23]. This bursting gives rise to broadly distributed firing rates (Fig 11B), where both populations display distributions with long exponentially

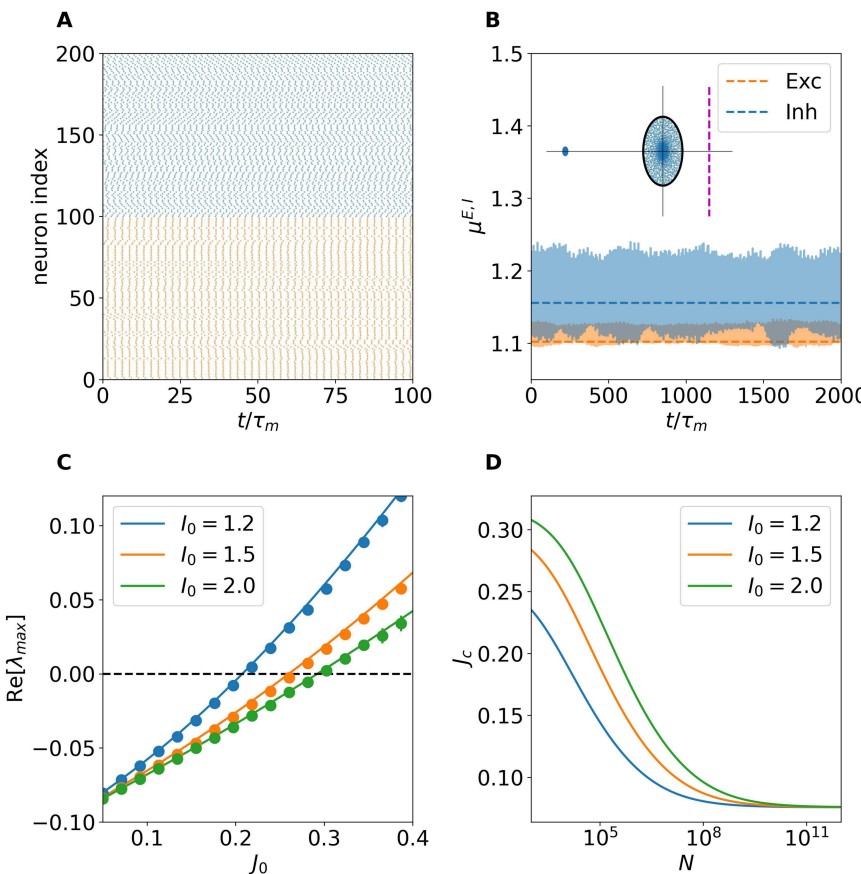

**Fig 10. Fixed point regime in the spiking network.** (A) Raster plot of a subset of excitatory (orange) and inhibitory (blue) neurons showing tonic firing patterns. (B) Sample synaptic input to two neurons in the excitatory and inhibitory populations (solid line) compared with the mean-field rate model prediction (dashed line). (C) Maximum real eigenvalue of the heterogeneous perturbation Jacobian $\mathbf{DF}_{het}$ (symbols) and the random matrix theory prediction (solid lines) for varying values of external input $I_0$. (D) Critical coupling strength $J_c$ estimated from the generalization of the Girko's law as a function of network size $N$. For this figure $J_0 = 0.1$, $I_0 = 1.2$, $N = 5,000$, $\tau_D = \tau_{syn} = 10\,\tau_m$.

decaying tails. The inset show the coefficient of variation (CV) distributions for excitatory and inhibitory neurons, showing a more irregular firing of inhibitory neurons, with mean values $CV^E \approx 1.43$ and $CV^I \approx 2.57$. This stronger burstiness in the inhibitory population can be understood by examining the total input currents of representative neurons in Fig 11C: excitatory neurons operate close to threshold ($\mu^E \sim 1$), while inhibitory neurons remain clearly sub-threshold ($\mu^I < 1$). Consequently, inhibitory activity is clearly fluctuation-driven. The neurons stay usually silent and are activated by large input fluctuations, whenever they are activated thanks to the slow decay of the synaptic current they remain supra-threshold for a time comparable with the synaptic time scale [23]. The panel also shows that the mean input currents predicted by the DMF rate model (dashed lines) match those obtained from the spiking network. Finally, Fig 11D compares the population-averaged autocorrelation functions: the DMF prediction (solid lines) exhibits excellent agreement with both the spiking network and the rate model, confirming that DMF theory faithfully captures the spiking network dynamics even when balance is maintained through nonlinear mechanisms such as short-term depression, and without the need for strong external drive.

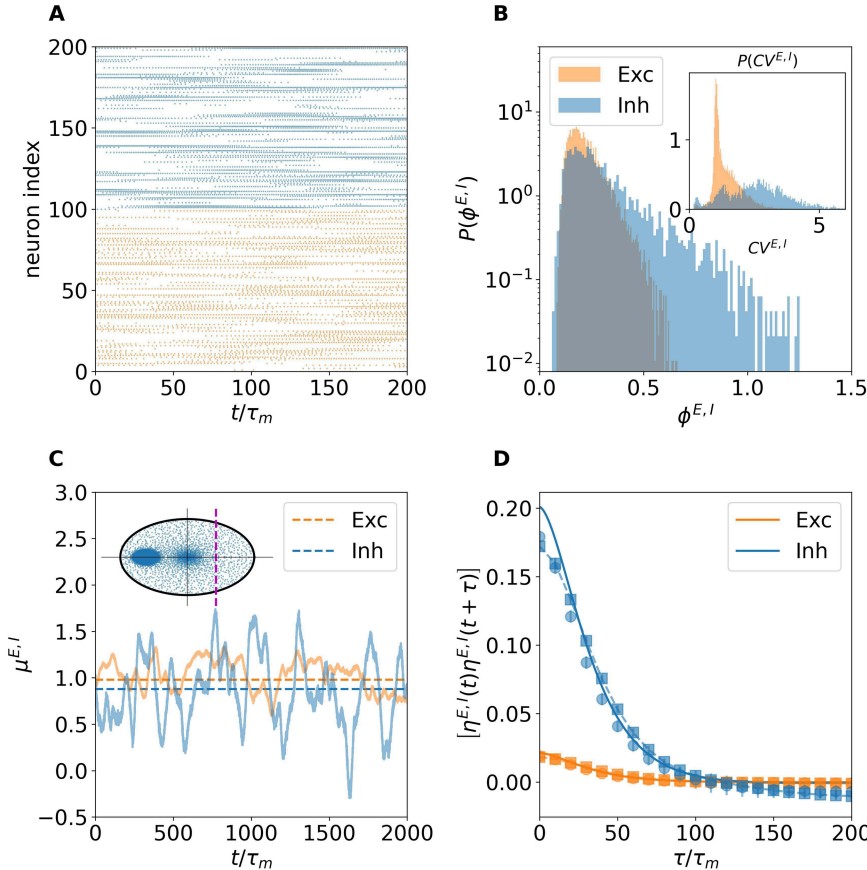

**Fig 11. Chaotic regime in the spiking network.** (A) Raster plot of a subset of excitatory (orange) and inhibitory (blue) neurons showing bursting firing patterns. (B) Distribution of firing rates (main panel) in the excitatory (orange) and inhibitory (blue) populations. the inset displays the distribution of the coefficient of variation $CV^{E,I}$. (C) Two samples of neuronal input from the spiking network (solid lines) and average calculated using rate model (dashed lines). (D) Input autocorrelation function for excitatory and inhibitory populations. Squares correspond to spiking network simulations, circles to rate model and solid lines to DMF predictions. For this figure $J_0 = 0.4$, $I_0 = 1.2$, $N = 5,000$, $\tau_D = \tau_{syn} = 10\,\tau_M$.

## Summary and discussion

We have carefully characterized, both numerically and theoretically, the balanced dynamics of densely connected excitatory–inhibitory networks composed of rate-based neurons. The balance originates from a novel mechanism relying on STD at excitatory–excitatory synapses, which is effective even under weak external external currents $\mathcal{O}(1)$ [17] at variance with classical mechanisms requiring strong external drives $\mathcal{O}(\sqrt{K})$ [5]. As discussed in Methods, this mechanism can already be understood by analyzing the stationary solutions for sufficiently large in-degrees $K \gg 1$. Namely, the partial input currents $h^{EE}$, $h^{EI}$ ($h^{IE}, h^{II}$) contributing to the total excitatory (inhibitory) input current $\mu^E$ ($\mu^I$) diverge as $\sqrt{K}$ for growing in-degrees. Therefore they should cancel each other to obtain input currents $\mu^{E,I} \simeq \mathcal{O}(1)$. Indeed we have found that the total input currents are always of the order of the external drive also at finite $N$ and for different values of $I_0$, as shown in the inset of Fig 2C for the stationary case as well as in Fig 5E for the chaotic regime. Finite $\mu^{E,I}$ in turn lead to finite values for the means and the fluctuations of the firing rates, as in the cortex. In the limit of large $K$, and in the absence of strong external currents, the stationary solutions for the rates can be found by solving a homogeneous system of equations. Nonzero stationary solutions can then be found only in the presence of nonlinear terms, with the required nonlinearity provided in the examined case by the synaptic depression acting on the excitatory-excitatory synapses. In S6 Appendix

we have shown that balanced states can emerge also in more complex scenarios, e.g. in presence of differential depression acting on excitatory-excitatory and inhibitory-inhibitory synapses. This result paves the way for a careful investigation on the role played by differential depression for the dynamics of balanced networks and for a comparison with experimental results obtained in the visual cortex along this direction [33]. However, this goes beyond the scope of our work and we leave it to future analysis.

The asymptotic solutions derived in the thermodynamic limit (6) reveal that firing rates remain finite and non-vanishing whenever the inhibitory current acting on the excitatory population is smaller than that acting on the inhibitory one. Moreover, these asymptotic firing rates are independent of the specific form of the transfer function. Instead, they depend only on the excitatory and inhibitory connectivity densities ($c_E$, $c_I$), the inhibitory gain factors ($g_E$, $g_I$), and the STD parameters. Finite size corrections to the asymptotic firing rates show, already at the first order in $1/\sqrt{N}$, a dependence on the remaining parameters of the model, as well as on the transfer function, as shown in S2 Appendix.

In finite systems, the dynamics is characterized by a homogeneous stable fixed point for sufficiently weak synaptic coupling ($J_0 \leq J_c$), a chaotic regime at strong coupling ($J_0 > J_r$), and a transition regime emerging at intermediate values of $J_c < J_0 < J_r$. We have shown that the homogeneous fixed point remains stable against homogeneous perturbations, but it loses stability under heterogeneous perturbations, leading to heterogeneous stationary or oscillatory solutions. The linear stability analysis was carried out by extending *Girko's circular law* to excitatory–inhibitory populations with STD, along the framework developed in [28].

The specific transition scenario towards rate chaos depends on the realization of the underlying random network, as we have numerically verified in selected cases by computing the largest Lyapunov exponents [36]. The transition regime always begins with the emergence of heterogeneous solutions—either stationary or periodic—triggered by the destabilization of the homogeneous fixed point. These solutions may then be followed by quasi-periodic dynamics or by alternating windows of stability and chaos, before eventually converging to fully developed rate chaos. This intermediate regime is extremely rich and certainly deserves a more detailed quantitative characterization, which lies beyond the scope of the present work and is left for future studies. Nevertheless, in S5 Appendix we provide an estimate of the scaling of the transition width $J_r - J_c$ with system size, showing that the width systematically shrinks as $N$ increases. This strongly suggests that, in the thermodynamic limit, the model exhibits an abrupt transition from a stable fixed point to rate chaos upon increasing the synaptic coupling. Analogous sharp transitions have been previously reported for rate models not following Dale's principle (i.e., with Gaussian-distributed synaptic couplings) [20], for rate models with classical balanced dynamics [21], and for spiking neural networks with sufficiently slow synaptic dynamics [22,23].

The rate chaos regime has been thoroughly characterized via numerical investigations combined with DMF techniques, developed by extending previous approaches for excitatory–inhibitory populations with frequency adaptation [31] to rate models with STD. For finite networks, we find that input current fluctuations increase with synaptic coupling and in a less dramatic fashion with the external drive. At the same time, the autocorrelation of the inputs decays more rapidly for increasing synaptic coupling, indicating faster loss of temporal correlations. As the system size grows, the influence of the external currents becomes negligible, while the mean firing rates of both excitatory and inhibitory populations, as well as the effective synaptic strength shaped by STD, stabilize to finite values. These findings clearly indicate that, also in the rate chaos regime, the novel balance mechanism is able to sustain finite firing activity, together with input currents with finite mean and fluctuations, even in the thermodynamic limit (see Figs 5 and 6 in Dynamic Mean Field Theory section). Furthermore, the asymptotic balanced regime is achieved for excitatory (inhibitory) in-degree $\simeq 10^4$ ($\simeq 10^3$), i.e. for values comparable with those measured in the cortex [10].

Moreover, we have analyzed in detail the influence of the network structure on the balancing mechanism. In our case, the network is densely connected, meaning that the in-degree $K$ grows proportionally with $N$. This class of networks has been investigated in [8] in the context of classically balanced networks. In particular, the authors in [8] demonstrated that correlations among partial input currents (either excitatory or inhibitory) persist even in large networks due to the presence of shared inputs. Nevertheless, the firing rates of the neurons and the net input currents remain essentially uncorrelated,

as in balanced sparse networks [6]. Similar results have also been reported experimentally for neuronal cultures [9]. We observe the same scenario in our model. Specifically, correlations among firing rates and net input currents vanish with increasing $N$, namely the corresponding Pearson correlation coefficients exhibit a power-law decay with an exponent $\gamma \simeq$ 0.7–0.8. Concurrently, the correlation coefficients of the partial input currents show a weak growth with the system size. Therefore, we can conclude that, analogously to [8], balance in this system is achieved through a dynamic cancellation of correlations driven by the tracking of fluctuations in the excitatory and inhibitory partial input currents, in agreement with *in vivo* observations [37,38]. Following the nomenclature introduced in [10], the present mechanism should be classified as *temporal balance*, since it operates on fast timescales. This in contrast with the *mean balance* observable in sparse random networks, where excitation and inhibition are balanced on average, but they may remain unbalanced at short times due to the lack of correlated fluctuations in partial excitatory and inhibitory input currents.

Finally, we have verified that the same scenario reported for rate models is also observable in LIF spiking neural networks with sufficiently slow synaptic dynamics, which can be well approximated by rate models with suitable transfer function. In particular, for spiking networks we find a stable homogeneous solution at weak synaptic coupling and a chaotic regime at strong coupling. The chaotic dynamics is fluctuation-driven, as neurons operate on average below or close to threshold. Overall, we observe a good agreement among numerical simulations of the LIF model and of the corresponding rate model as well as with the DMF predictions.

Experimental investigations of neural cultures [9] confirmed the main assumptions and results predicted for classically balanced regimes [5] in densely connected networks [8]. In particular, [9] showed that the amplitude of excitatory and inhibitory PSPs decreases as $\sim 1/\sqrt{K}$ with the mean connectivity $K$, thus confirming one of the central predictions of balanced theory. Moreover, by optogenetic stimulation of a subset of neurons they demonstrated that excitation and inhibition are tightly balanced, that fluctuation amplitudes remain independent of $K$, that neuronal firing is essentially Poisson-like, and that balance arises through excitatory–inhibitory tracking. All these observations are consistent with the theory of balance extended to densely connected recurrent networks [8]. Remarkably, these results were obtained far from the asymptotic regime where both $K$ and $N$ are very large; in fact, the total connectivity in the experiments ranged only from 80 to 600.

Some aspects of the analysis in [9] deserve further discussion for their implications on our approach. In [6] it was shown that, under the assumption of strong external currents, scaling as $I_0 = \sqrt{K} i_0$, the mean firing rates of neurons in balanced networks should grow linearly with $i_0$. To test this prediction, [9] stimulated a subset $M_S$ of neurons with light pulses of amplitude $J$ delivered at frequency $\nu_S$. The external current in this setup is proportional to $M_S J K \nu_S$, since each stimulated neuron projects to $K$ post-synaptic targets. To ensure that the input current scales as $\sqrt{K}$, the number of stimulated neurons $M_S$ was accordingly rescaled, thus yielding $I_0 = \sqrt{K} i_0 = \sqrt{K}(M_S J \nu_S)$. The authors then varied $i_0$ either by changing $M_S$ or by adjusting $\nu_S$. In the first case, firing rates grew linearly with $M_S$, whereas in the second case they saturated for large $\nu_S$. This saturation persisted even after correcting for frequency-dependent declines in channelrhodopsin efficacy, leading the authors to suggest that *synaptic depression and/or firing rate adaptation may also contribute to the saturation*. In summary, while the study assumed strong external currents, it did not directly test their strength, and it also provided hints that nonlinear adaptive processes could play a role in the balancing mechanism.

The assumptions underlying classical balanced theory [5,8] were recently re-examined in [10]. There, the authors presented experimental evidences suggesting that cortical circuits often operate in a so-called *loosely balanced regime*, where partial, net, and external input currents are all of order $\mathcal{O}(1)$. In this regime, balance is not achieved through the cancellation of large excitatory, inhibitory, and external drives of order $\mathcal{O}(\sqrt{K})$, but rather through weaker inputs with comparable magnitudes. Importantly, even in the loosely balanced regime, the net input remains of the same order as the distance to threshold, so neuronal activity remains fluctuation-driven, producing the irregular spiking observed in cortex. Large part of these results concern sensory cortices, leaving open the possibility that tight balance may still dominate in areas with much higher connectivity (where $K \sim 5,000$), such as the frontal cortex.

Inspired by these criticisms, Abbott and collaborators recently proposed a novel balance mechanism termed *sparse balance* [11], which combines weak external currents with broadly distributed synaptic weights. In this scenario, balance produces nonlinear population responses to uniform inputs (responses that are expected in cortical circuits [10,39] but absent in the tight balanced regime) and input currents with non-Gaussian statistics. However, sparse balance comes at the cost that the fraction of active neurons decays as $1/\sqrt{K}$, vanishing in the thermodynamic limit.

As already mentioned in the Introduction, strong external currents follows from assuming for the feed-forward connections an out-degree $\mathcal{O}(K)$ and synaptic strengths $\mathcal{O}(1/\sqrt{K})$. If the external current is weak, one of this two hypotheses should be not true. Most of the experimental evidences showing that input currents are $\mathcal{O}(1)$ involve thalamo-cortical connections. In particular, it has been shown that the thalamic net excitation to the mice visual cortex, with optogenetic silencing of intra-cortical excitatory input, induce a depolarization that is of the order of the typical distance to threshold [10,14,15]. This implies that the feed-forward input is $\mathcal{O}(1)$ [11]. Evidences that the out-degree from thalamus to cortex is smaller than in the cortico-thalamo-cortical loop have been reported for macaque in [40]. Quite recently the number of thalamic neurons converging to a cortical neuron in the primary visual cortex of mouse has been estimated giving out-degrees ranging from to 2-7 and accounting for the 90% of the total synaptic weight to the cortical neuron [41]. This result should be compared with a previous study indicating a mean out-degree of $\simeq 80$ for thalamic inputs into layer 4 neurons [42]. Apart the exact value of the out-degree that can be affected by the method employed to estimate it (for a detailed discussion see [41]) it is clear that the feed-forward thalamo-cortical connections are definitely sparser than in the cortex. This seems to suggest that the hypothesis that the out-degree is $\mathcal{O}(K)$ is not valid, at least in this specific case, and this could be at the origin of the measured $\mathcal{O}(1)$ external currents.

Taken together, the experimental results of [9] and the ones reported in [10] make it plausible that the balance observed in brain circuits may rely on strong synapses (scaling as $1/\sqrt{K}$), but not necessarily on strong external currents, and that instead it may be stabilized by nonlinear mechanisms affecting neuronal firing, such as STD, as we have demonstrated here and in spiking networks [17]. Other adaptive processes may also play similar roles. For instance, spike-frequency adaptation has been shown to restore balance in networks with highly heterogeneous connectivity [43], while facilitation can promote bistability in balanced regimes [44]. However, both these studies assumed strong external inputs. Extending the present analysis to other forms of biologically relevant short-term and long-term plasticity under weak inputs $\mathcal{O}(1)$ is an important direction for future investigation.

Despite the rich finite-size phenomenology observed in our model, both Lyapunov analyses and DMF predictions indicate that in the thermodynamic limit the transition from the stable homogeneous fixed point to the chaotic regime becomes sharp. This is precisely the scenario firstly reported in [20] for fully coupled rate networks with Gaussian-distributed couplings, and later extended to randomly diluted networks in the balanced state with strong inputs [21], and to spiking neural networks with slow synaptic dynamics [22,23]. In [21,22] general DMF equations were derived for excitatory–inhibitory populations, though solved only for purely inhibitory networks with external drive. It was only later that Mastrogiuseppe & Ostojic [28] solved the DMF equations for excitatory–inhibitory networks without external input. In that case, the inclusion of excitatory populations induced a strong increase in firing rates, even diverging at sufficiently large coupling, thus requiring a saturation mechanism such as STD to stabilize activity, exactly as in our model. Accordingly, we have shown that the bifurcation scenario described in [20] extends also to excitatory–inhibitory populations balanced through short-term depression with weak inputs $\mathcal{O}(1)$.

The bifurcation structure can be significantly modified if neurons and synapses evolve under Hebbian or anti-Hebbian long-term plasticity rules [35]. DMF analysis has revealed that slow synaptic modifications can reshape the phase diagram, delaying chaos or even generating oscillatory modes [35]. A natural extension of our work would therefore be to study sparse excitatory–inhibitory networks balanced through short-term plasticity while simultaneously undergoing long-term synaptic evolution, in order to understand how these two adaptive processes interact to shape network dynamics.

Moreover, for single-population Gaussian random networks it has been shown that rate chaos is extensive, i.e. the number of positive Lyapunov exponents grows with $N$ [45,46]. In our model, chaos is typically hyperchaotic, with at least

two positive Lyapunov exponents (see Fig 9 in Routes to Rate Chaos). A full characterization of the Lyapunov spectrum would be an interesting future step, to assess the role of STD in modulating chaos extensivity.

The role played by current-based (linear) and conductance-based (nonlinear) synapses for the emergence of asynchronous irregular dynamics has been previously examined in [47]. In particular the authors have shown that while for linear synapses a small finite external current is needed to self-sustain the asynchronous irregular activity [1], this is not the case for conductance-based synapses. In this latter case the firing activity can be sustained for a finite time period even in the absence of external drive. Furthermore, the authors have shown that the life-time of this state diverges with the system size. These results show that a different type of nonlinearity in the neuronal response (the one associated to conductance-based mechanisms) can lead to findings that strongly resonate with our results. Furthermore, the investigation of conductance-based neurons has recently revealed [48] that in these models irregular dynamics can emerge dynamically, without finite tuning, whenever the synaptic strengths scale as $1/\log(K)$. Therefore to observe a self-sustained asynchronous firing activity definitely stronger synapses are required for conductance-based neurons than for current-based ones, where the synaptic strength are usually assumed to be $\mathcal{O}(1/\sqrt{K})$.

These findings together with our analysis indicate that nonlinear mechanisms can lead to completely new scenarios in the context of dynamically balanced theory, our paper presents one of these possible scenarios. However, further investigations are required to fully understand the role played by nonlinear responses on promoting self-sustained asynchronous irregular activity in excitatory-inhibitory cortical circuits.

In conclusion, our results demonstrate that a biologically grounded synaptic nonlinearity—short-term depression—can resolve a fundamental limitation of classical balanced network theory, the need for strong external drive. By combining DMF analysis with large-scale simulations, we have shown that depression-stabilized balance is robust and general: it persists across neuronal nonlinearities, model parameterizations, and network sizes, while reproducing cortical-like statistics and variability of firing activity. This work thus bridges theoretical models of balanced chaos with experimentally observed synaptic dynamics, providing new insight into how cortical circuits may sustain irregular yet stable activity. More broadly, our study highlights the importance of incorporating realistic synaptic dynamics into network theories, thereby advancing our understanding of excitation–inhibition balance in the brain.

## Methods

### Parameters of the network model

In Table 1 we report the values of the parameters employed throughout this work unless otherwise stated.

### Homogeneous stationary solution

In the following we report the linear stability analysis of homogeneous stationary solutions subject to homogeneous and heterogeneous perturbations.

**Linear stability analysis for homogeneous perturbations.** In order to analyze the stability of the stationary homogeneous solutions (5) to homogeneous perturbations we consider the following mean-field formulation of the network model (14):

$$\dot{x}_m^E = -x_m^E + J_0 j_E(\sqrt{K_E}\phi[x_m^E]w_0 - \sqrt{K_I}g_E\phi[x_m^I]) + I_0 \tag{14a}$$

$$\dot{x}_m^I = -x_m^I + J_0 j_I(\sqrt{K_E}\phi[x_m^E] - \sqrt{K_I}g_I\phi[x_m^I]) + I_0 \tag{14b}$$

$$\dot{w}_m = \frac{1-w_m}{\tau_D} - uw_m\phi[x_m^E]; \tag{14c}$$

**Table 1. Values of the employed parameters.**

| Parameter | Value | Parameter | Value |
|-----------|-------|-----------|-------|
| $N$ | 20000 | $\tau_D$ | 10 |
| $f$ | 0.8 | $g_E$ | 1 |
| $c_E$ | 0.025 | $g_I$ | 2 |
| $c_I$ | 0.005 | $j_E$ | 1 |
| $u$ | 0.5 | $j_I$ | 1.5 |

and the associated eigenvalue problem:

$$|\mathbf{DF}_{\text{hom}} - \lambda \mathbf{I}| = 0.$$

Here $|\cdot|$ denotes the determinant of a matrix, $\lambda$ are the -possibly complex- eigenvalues and $\mathbf{I}$ is the identity matrix. The Jacobian matrix of the mean-field homogeneous system (14) evaluated at the homogeneous fixed point $(x_0^E, x_0^I, w_0)$ is given by

$$\mathbf{DF}_{\text{hom}} = \begin{pmatrix} -1 + J_0\sqrt{K_E}j_E\phi_E'w & -J_0\sqrt{K_I}g_Ej_E\phi_I' & J_0\sqrt{K_E}j_E\phi^E \\ J_0\sqrt{K_E}\phi_E'j_I & -1 + J_0\sqrt{K_I}g_Ij_I\phi_I' & 0 \\ -uw\phi_E' & 0 & -(\tau_D^{-1} + u\phi^E) \end{pmatrix}. \tag{15}$$

Here we have made use of the short-hand notation $\phi^\alpha \equiv \phi[x_0^\alpha]$ and similarly for $\phi_\alpha' \equiv \phi'[x_0^\alpha]$.

The characteristic polynomial of the eigenvalue problem is in this case of the third order and it can be written as $p_{hom}(\lambda) = \lambda^3 + \beta_2\lambda^2 + \beta_1\lambda + \beta_0$ with

$$\beta_0 = -det(\mathbf{DF}_{\text{hom}}) \tag{16}$$

$$\beta_1 = \sum_i M_{i,i} \tag{17}$$

$$\beta_2 = -Tr(\mathbf{DF}_{\text{hom}}) \tag{18}$$

where $M_{i,j}$ is the minor resulting from the deletion of the $i$-th row and $j$-th column. Taking into consideration the Routh-Hurwitz stability criterion, the real part of all the eigenvalues will be negative iif $\beta_i > 0 \; \forall i$ and $\beta_2\beta_1 > \beta_0$.

**Linear stability analysis for heterogeneous perturbations.** The eigenvalues derived in the previous sub-section only describe the linear stability of the homogeneous stationary solution for the very special case of homogeneous perturbations, however in general the perturbations are heterogeneous. In this case the Jacobian takes the form

$$\mathbf{DF}_{\text{het}} = \left( \begin{array}{cc|c} -\mathbf{I} + \mathbf{J}^{EE}J_0\phi_E'w & \mathbf{J}^{EI}J_0\phi_I' & \mathbf{J}^{EE}J_0\phi^E \\ \mathbf{J}^{IE}J_0\phi_E' & -\mathbf{I} + \mathbf{J}^{II}J_0\phi_I' & \mathbf{0} \\ \hline -\mathbf{I}uw\phi_E' & \mathbf{0} & -\mathbf{I}(\tau_D^{-1} + u\phi^E) \end{array} \right) \tag{19}$$

By recalling that for a block matrix of the form:

$$\mathbf{Z} = \begin{pmatrix} \mathbf{A} & \mathbf{B} \\ \mathbf{C} & \mathbf{D} \end{pmatrix};$$

if the matrix $\mathbf{D}$ is invertible, then the determinant of the matrix $\mathbf{Z}$ is given by

$$|\mathbf{Z}| = |\mathbf{A} - \mathbf{B}\mathbf{D}^{-1}\mathbf{C}| \cdot |\mathbf{D}|. \tag{20}$$

In order to solve the eigenvalue problem for the heterogeneous case, we set $\mathbf{Z} = \mathbf{DF}_{\text{het}} - \lambda\mathbf{I}$ and apply the identity (20). Finally, the eigenvalues of $\mathbf{DF}_{\text{het}}$ can be obtained by solving the following equations:

$$\left| \begin{array}{cc} \mathbf{J}^{EE}J_0\phi'_E w\left(1 + \frac{u\phi^E}{\tau_D^{-1} + u\phi^E + \lambda}\right) - (1+\lambda)\mathbf{I} & \mathbf{J}^{EI}J_0\phi'_I \\ \mathbf{J}^{IE}J_0\phi'_E & \mathbf{J}^{II}J_0\phi'_I - (1+\lambda)\mathbf{I} \end{array} \right| \cdot \left|-(\tau_D^{-1} + u\phi^E)\mathbf{I} - \lambda\mathbf{I}\right| = 0. \tag{21}$$

The first determinant includes a nonlinear dependence on $\lambda$, which makes obtaining a closed-form analytic solution rather difficult. To proceed, we introduce a zeroth-order approximation in which the $\lambda$ term inside the nonlinear contribution is neglected. This simplification is expected to be accurate in the vicinity of the bifurcations, where $\lambda$ is either real or have small imaginary parts and crosses the imaginary axis. Under this approximation, the eigenvalue problem associated with the left-most determinant in Eq (21) reduces to the following form:

$$|\mathbf{P} - \lambda_P\mathbf{I}| \equiv \left| J_0 \left( \begin{array}{cc} a\mathbf{J}^{EE} & b\mathbf{J}^{EI} \\ c\mathbf{J}^{IE} & b\mathbf{J}^{II} \end{array} \right) - \lambda_P\mathbf{I} \right| = 0 \tag{22}$$

where

$$a = \phi'_E w\left(1 + \frac{u\phi^E}{\tau_D^{-1} + u\phi^E}\right), \; b = \phi'_I, \; c = \phi'_E \tag{23}$$

and $\lambda_P = (1 + \lambda)$ are the eigenvalues of the approximated block matrix $\mathbf{P}$.

The matrix $\mathbf{P}$ is random but contains cell-type-specific sparse connectivity, for this kind of random matrix a series of papers [25–27] have generalized the classical Girko's law [24] to obtain the distribution of the corresponding eigenvalues. In particular, by following [28] we can affirm that the eigenspectrum of the matrix $\mathbf{P}$ is composed of a continuous part, lying within a complex circle centered in $(-1,0)$, and a discrete part, made of two outliers eigenvalues. The radius of the complex circle is determined by the square root of the largest eigenvalue of the $2 \times 2$ matrix $\Sigma$ containing the variances of the entries distributions in the four blocks multiplied by $N$:

$$\Sigma = J_0^2 \left( \begin{array}{cc} a^2 j_E^2 & b^2 g_E^2 j_E^2 \\ c^2 j_I^2 & b^2 g_I^2 j_I^2 \end{array} \right). \tag{24}$$

The radius of the circle in this case is then:

$$r = \frac{J_0}{\sqrt{2}}\sqrt{(a^2 j_E^2 + b^2 g_I^2 j_I^2) + \sqrt{(a^2 j_E^2 + b^2 g_I^2 j_I^2)^2 + 4b^2 j_E^2 j_I^2(c^2 g_E^2 - a^2 g_I^2)}}. \tag{25}$$

The two outliers are the eigenvalues of $2 \times 2$ matrix $\mathbf{M}$, obtained by estimating the mean of the matrix $\mathbf{P}$ in each of the four blocks multiplied by $N$:

$$\mathbf{M} = J_0 \left( \begin{array}{cc} a j_E \sqrt{K_E} & -b g_E j_E \sqrt{K_I} \\ c j_I \sqrt{K_E} & -b g_I j_I \sqrt{K_I} \end{array} \right) \tag{26}$$

The outliers are thus given by

$$\lambda_{out}^{\pm} = \frac{J_0}{2} \left\{ (aj_E\sqrt{K_E} - bg_Ij_I\sqrt{K_I}) \pm \sqrt{(aj_E\sqrt{K_E} - bg_Ij_I\sqrt{K_I})^2 + 4bj_Ej_I\sqrt{K_EK_I}(ag_I - cg_E)} \right\}. \tag{27}$$

The whole spectrum is completed by the eigenvalues given by the second determinant in Eq (21):

$$|\mathbf{Q} - \lambda_Q\mathbf{I}| \equiv \left| -(\tau_D^{-1} + u\phi^E)\mathbf{I} - \lambda_Q\mathbf{I} \right| = 0 \tag{28}$$

which are simply given by:

$$\lambda_Q = -(\tau_D^{-1} + u\phi^E). \tag{29}$$

It is important to note that the system cannot lose stability through the eigenvalues given in Eq (29), as these remain strictly negative across the parameter space.

Consequently, we must focus on the eigenvalues of the matrix $\mathbf{P}$, which determine the onset of instability. We have numerically verified, by directly diagonalizing the *approximated* matrix $\mathbf{P}$ defined in Eq (22), that the dense part of the spectrum is indeed contained within the circle of radius (25), and that the outliers are accurately described by Eq (27). However, numerical diagonalization of the full Jacobian $\mathbf{DF}_{het}$ reveals that, while the bulk of the spectrum is well captured by the generalized *Girko's criterion* with radius given by (25), the outliers are not equally well reproduced by the approximated matrix $\mathbf{P}$. In particular, as shown in S3 Appendix, the smaller outlier is reasonably well approximated by the expression $\lambda_{out}^-$; it follows the same scaling law with $N$, and its real part becomes increasingly negative, thus not contributing to the onset of instability. In contrast, the larger outlier does not scale in the same way as $\lambda_{out}^+$ (see S3 Appendix), which is expected to grow with $N$. According to our analysis, the largest eigenvalue of the full Jacobian remains inside Girko's circle at least near the transition meaning that in the real system, the transition is governed exclusively by the crossing of the imaginary axis by the eigenvalues lying within the circle of radius $r$ (similarly to what was recently reported in [35]). This discrepancy between the approximated matrix and the full Jacobian is likely a consequence of the zeroth-order approximation used in deriving the simplified form of $\mathbf{P}$. Therefore, the condition to determine the critical value of the synaptic coupling $J_c$ that leads to the instability of the homogeneous stationary solution can be expressed as:

$$1 = \frac{J_c^2}{2} \left[ (a^2j_E^2 + b^2g_I^2j_I^2) + \sqrt{(a^2j_E^2 + b^2g_I^2j_I^2)^2 + 4b^2j_E^2j_I^2(c^2g_E^2 - a^2g_I^2)} \right], \tag{30}$$

where all other parameter values are fixed.

## Dynamic mean field theory

Once the chaotic regime establishes we can characterize the dynamics via a generalization of the DMF theory [20,28]. In particular, we rewrite the dynamical evolution of the coupled firing rate models Eq (6) as a set of $N$ Langevin equations for the excitatory and inhibitory neurons, while the evolution of the STD variables $\{w_i\}$ remain deterministic, namely

$$\dot{x}_i^E = -x_i^E + \eta_i^E \tag{31a}$$

$$\dot{x}_i^I = -x_i^I + \eta_i^I \tag{31b}$$

$$\dot{w}_i = \frac{1 - w_i}{\tau_D} - uw_i\phi[x_i^E] \tag{31c}$$

where the deterministic input currents $\{\mu_i^E, \mu_i^I\}$ have been replaced by effective Gaussian noise terms $\{\eta_i^E, \eta_i^I\}$, which are therefore completely characterized by their first and second moments.

In order to estimate self-consistently these moments in the large $N$ limit, we substitute the averages on neurons, initial conditions and network realizations (denoted as $\langle \cdot \rangle$) with averages over the realization of the stochastic processes (denoted as $[\cdot]$). In this set-up the the first moments read as:

$$[\eta_i^E] = [\eta^E] = \left[ \sum_{j \in E}^{N_E} J_{ij}^{EE} \phi[x_j^E] w_j + \sum_{j \in I}^{N_I} J_{ij}^{EI} \phi[x_j^I] + I_0 \right] \tag{32}$$

$$= J_0 j_E \sqrt{N} \left( \sqrt{c_E} \tilde{r}_E - g_E \sqrt{c_I} r_I \right) + I_0 \tag{33}$$

$$[\eta_i^I] = [\eta^I] = \left[ \sum_{j \in E}^{N_E} J_{ij}^{IE} \phi[x_j^E] + \sum_{j \in I}^{N_I} J_{ij}^{II} \phi[x_j^I] + I_0 \right] \tag{34}$$

$$= J_0 j_I \sqrt{N} \left( \sqrt{c_E} r_E - g_I \sqrt{c_I} r_I \right) + I_0 \tag{35}$$

where $\tilde{r}_E \equiv \langle \phi_i^E w_i \rangle = [\phi^E w]$, $r_E \equiv \langle \phi_i^E \rangle = [\phi^E]$ and $r_I \equiv \langle \phi_i^I \rangle = [\phi^I]$.

The second moments correspond to the following noise autocorrelations:

$$[(\eta_i^E(t) - [\eta^E])(\eta_i^E(t + \tau) - [\eta^E])] = J_0^2 j_E^2 \left\{ \tilde{C}^E(\tau) - \tilde{r}_E^2 + g_E^2 (C^I(\tau) - r_I^2) \right\} \tag{36a}$$

$$[(\eta_i^I(t) - [\eta^I])(\eta_i^I(t + \tau) - [\eta^I])] = J_0^2 j_I^2 \left\{ C^E(\tau) - r_E^2 + g_I^2 (C^I(\tau) - r_I^2) \right\} \tag{36b}$$

where $C^{E,I}(\tau) = \langle \phi_i^{E,I}(t) \phi_i^{E,I}(t + \tau) \rangle$ and $\tilde{C}^E = \langle \phi_i^E(t) w_i(t) \phi_i^E(t + \tau) w_i(t + \tau) \rangle$ are the rate auto-correlation functions. In Eqs (32) and (36) we have used the fact that for sufficiently large $N$ firing rates $\{\phi[x_i^E]\}$ and $\{\phi[x_i^I]\}$ as well as the total input currents $\{\mu_i^E\}$ and $\{\mu_i^I\}$ behave independently and therefore the Langevin equations (31a) and (31b) completely decouple in the thermodynamic limit $N \to \infty$. This independence property for large $N$ also means that the cross-correlations of the noise $\left[ (\eta_i^E(t) - [\eta^E])(\eta_j^E(t + \tau) - [\eta^E]) \right]$ vanish as shown in the subsection *Balancing mechanism in densely connected recurrent networks*.

The mean field currents $x_i^{E,I}$ obtained by the integration of the Eqs (31a) and (31b), corresponding to Orstein-Uhlenbeck processes, are Gaussian variables with finite time correlation. Therefore they can be completely characterized in terms of their first two moments, the first ones are given by

$$\mu^E = [x_i^E] = [\eta^E] \tag{37}$$

$$\mu^I = [x_i^I] = [\eta^I] \tag{38}$$

while for the corresponding mean-subtracted correlation functions we have

$$\Delta^{E,I}(\tau) = [x_i^{E,I}(t) x_i^{E,I}(t + \tau)] - [x_i^{E,I}]^2. \tag{39}$$

The evolution equations for $\Delta^{E,I}(\tau)$ are obtained differentiating twice w.r.t $\tau$, leading to the second order differential equation

$$\frac{d^2 \Delta^{E,I}(\tau)}{d\tau^2} = \Delta^{E,I}(\tau) - [(\eta_i^{E,I}(t) - [\eta^{E,I}])(\eta_i^{E,I}(t + \tau) - [\eta^{E,I}])]. \tag{40}$$

By setting $\Delta^{E,I}(\tau = 0) = \Delta_0^{E,I}$ and by noticing that $\Delta^{E,I}(\tau = \infty) = 0$ due to the homogeneity of the in-degree distribution, we finally arrive to the self-consistent equations for the mean and the variances of the input currents, namely

$$\mu^E = J_0 j_E \sqrt{N} (\sqrt{c_E} [\phi^E w] - g_E \sqrt{c_I} [\phi^I]) + I_0 \tag{41a}$$

$$\mu^I = J_0 j_I \sqrt{N} \left( \sqrt{c_E} [\phi^E] - g_I \sqrt{c_I} [\phi^I] \right) + I_0 \tag{41b}$$

$$\Delta_0^E = J_0^2 j_E^2 \left\{ [(\phi^E w)^2] - [\phi^E w]^2) + g_E^2 ([(\phi^I)^2] - [\phi^I]^2) \right\} \tag{42a}$$

$$\Delta_0^I = J_0^2 j_I^2 \left\{ [(\phi^E)^2] - [\phi^E]^2 + g_I^2 ([(\phi^I)^2] - [\phi^I]^2) \right\}. \tag{42b}$$

In these equations the dependence by single neuron index $i$ is finally dropped, since due to the statistical equivalence of the neurons within each population we can now represent the mean-field dynamics as two single-site Langevin equations one for each population plus one equation for the synaptic efficacy. This system of three equations has been previously reported in (10).

The quantities $[\phi]$ and $C(\tau)$ can be estimated as integrals over the Gaussian distributions

$$[\phi^{E,I}] = \int \mathcal{D}z \ \phi \left( \mu^{E,I} + \sqrt{\Delta_0^{E,I}} z \right) \tag{43a}$$

$$[\phi^E w] = \int \mathcal{D}z \ \phi \left( \mu^E + \sqrt{\Delta_0^E} z \right) \ w(z) \tag{43b}$$

and

$$C^{E,I}(\tau) = \int \mathcal{D}z \ \left\{ \int \mathcal{D}x \ \phi^{E,I} \left( \mu^{E,I} + \sqrt{\Delta_0^{E,I} - |\Delta^{E,I}(\tau)|} x + \sqrt{|\Delta^{E,I}(\tau)|} z \right) \right\}^2 \tag{44a}$$

$$\tilde{C}^E(\tau) = \int \mathcal{D}z \ \left\{ \int \mathcal{D}x \ \phi^E \left( \mu^E + \xi(x,z) \right) w(\xi(x,z)) \right\}^2 \tag{44b}$$

$$\xi(x,z) = +\sqrt{\Delta_0^E - |\Delta^E(\tau)|} x + \sqrt{|\Delta^E(\tau)|} z \tag{44c}$$

where $x$ and $z$ are Gaussian variables with zero mean and unitary variance. Here the notation $\int \mathcal{D}z = \int_{-\infty}^{\infty} dz e^{-z^2/2}/\sqrt{2\pi}$.

The equations (41), (42) and (43) describe quantities that evolves chaotically in time subject to temporal fluctuations, wherein $w(\cdot)$ reflects the fact that $w$ is a stochastic variable with non-trivial dependence on the Gaussian variable $z$. As this functional dependence is unknown we rely on numerical estimations of the autocorrelation functions, as explained in the following.

**Numerical estimation of the noise auto-correlation functions**

Solving self-consistently Eqs (36), (40), (41), (42), (43) and (44) to obtain the noise auto-correlation function in the present case is prohibitive. In practice, we solve iteratively these equations by following the procedure proposed in [31], which resembles the DMF methods employed to obtain the spectra associated to the firing activity in spiking neural networks [49,50].

This procedure recursively updates the mean input currents and the power spectral densities (PSDs) of the fluctuations of the total input currents for both populations. At each iteration, stochastic realizations of the colored Gaussian noise for excitatory and inhibitory neurons are generated accordingly to a PSD $S^{E,I}(\omega)$ defined in the frequency domain $\omega$ as follows

$$\eta^{E,I}(t) = \mathcal{F}^{-1} \left[ e^{i\theta} \sqrt{2 S^{E,I}(\omega)} \right],$$

where $\mathcal{F}^{-1}$ denotes an inverse Fourier transform and $\theta$ are random phases uniformly distributed in $[0, 2\pi]$. We set the initial PSD to be a flat spectrum (white noise).

Given these noise realizations, the mean field input variables $x^E(t)$, $x^I(t)$ and depression variable $w(t)$ are evolved by following the equations (10) integrated by employing an Euler scheme with a time step $dt = 0.06$. After an initial transient of duration $T_t = 1,000$, the empirical single-site average firing rates $r_E = \overline{\phi(x_E(t))}$, $r_I = \overline{\phi(x_I(t))}$, and $\tilde{r} = \overline{w(t)\phi(x_E(t))}$ are estimated by averaging over a time interval $T_a = 200,000$.

This leads to the estimation of the average input currents for the present iteration, namely:

$$\hat{\mu}^E = J_0 j_E \sqrt{N}(\sqrt{c_E}\tilde{r} - g_E\sqrt{c_I}r_I) + I_0 \tag{45}$$

$$\hat{\mu}^I = J_0 j_I \sqrt{N}(\sqrt{c_E}r_E - g_I\sqrt{c_I}r_I) + I_0, \tag{46}$$

and of the spectra associated to the current fluctuations

$$S_\beta(\omega) = |\mathcal{F}[\beta(t) - \langle\beta\rangle]|^2, \tag{47}$$

where $\beta = \{r_E, r_I, \tilde{r}\}$ and $\mathcal{F}$ is the direct Fourier transform.

These are then used to update the mean inputs and the spectra of the effective noise to be employed in the next iteration:

$$\mu^E \leftarrow (1 - \alpha)\mu^E + \alpha\hat{\mu}^E,$$
$$\mu^I \leftarrow (1 - \alpha)\mu^I + \alpha\hat{\mu}^I,$$
$$S^E(\omega) \leftarrow (1 - \alpha)S^E(\omega) + \alpha J_0^2 j_E^2 \left[S_{\tilde{r}}(\omega) + g_E^2 S_{r_I}(\omega)\right],$$
$$S^I(\omega) \leftarrow (1 - \alpha)S^I(\omega) + \alpha J_0^2 j_I^2 \left[S_{r_E}(\omega) + g_I^2 S_{r_I}(\omega)\right],$$

where the parameter $\alpha < 1$ is employed to avoid instabilities in the iterations and to guarantee the convergence the method, as similarly done also in [49,50]. The parameter $\alpha$ needs to be tuned depending on the set system size and the other parameters of the model. For example, for the analysis at $N = 5,000$ a factor in the range $0.1 \leq \alpha \leq 0.4$ guarantees the convergence of the algorithm, instead at $N = 10^{10}$ the parameter value should be set to $\alpha = 10^{-4}$.

Once the convergence is reached (usually 500 iterations are sufficient) the noise auto-correlation functions are obtained via the inverse Fourier transform of the final spectra:

$$[\eta^E(t)\eta^E(t + \tau)] = \mathcal{F}^{-1}[S^E(\omega)], \quad [\eta^I(t)\eta^I(t + \tau)] = \mathcal{F}^{-1}[S^I(\omega)].$$

The number of Fourier modes employed to estimate the Fourier transforms is usually fixed to 16,384.

## Correlation coefficients of the input currents and of the firing rates

To assess whether a dynamic cancellation mechanism—similar to the one described in [8]—is also active in our case, despite the different nature of the underlying balance, we introduce a set of indicators to quantify the level of correlation among the partial and total synaptic inputs, as well as among the firing activities of the neurons.

Let us first define the instantaneous partial input currents

$$h_i^{EE}(t) = \sum_{j\in E}^{N_E} J_{ij}^{EE}\phi[x_j^E(t)]w_j(t) + I_0 \qquad h_i^{EI}(t) = \sum_{j\in I}^{N_I} J_{ij}^{EI}\phi[x_j^I(t)], \tag{48a}$$

$$h_i^{IE}(t) = \sum_{j\in E}^{N_E} J_{ij}^{IE}\phi[x_j^E(t)] + I_0 \qquad h_i^{II}(t) = \sum_{j\in I}^{N_I} J_{ij}^{II}\phi[x_j^I(t)] \;; \tag{48b}$$

The constant input current $I_0$ has been incorporated into the excitatory input currents, as it is positive. However, it remains irrelevant for the analysis of input correlations, since it is both constant and of order $\mathcal{O}(1)$. It is straightforward to see that the total input currents $\mu_i^E(t)$ ($\mu_i^I(t)$) received by the $i$-th excitatory (inhibitory) neuron are given by

$$\mu_i^E(t) = h_i^{EE}(t) + h_i^{EI}(t) \qquad , \qquad \mu_i^I(t) = h_i^{IE}(t) + h_i^{II}(t). \tag{49}$$

Furthermore, we denote the variances of the partial input currents impinging neuron $i$ defined in (48) as $\Delta_i^{EE}$, $\Delta_i^{EI}$, $\Delta_i^{IE}, \Delta_i^{EE}$. Those associated to the total excitatory and inhibitory input currents are instead denoted as $\Delta_i^E$ and $\Delta_j^I$, respectively.

The level of correlations among the input currents are measured in terms of their population averaged correlation coefficients, namely

$$\rho^{EE} = \frac{1}{N_E^2}\sum_{i,j\in E}^{N_E}\frac{\overline{\delta h_i^{EE}(t)\delta h_j^{EE}(t)}}{\sqrt{\Delta_i^{EE}\Delta_j^{EE}}} \qquad \rho^{EI} = \frac{1}{N_I^2}\sum_{i,j\in I}^{N_I}\frac{\overline{\delta h_i^{EI}(t)\delta h_j^{EI}(t)}}{\sqrt{\Delta_i^{EI}\Delta_j^{EI}}}, \tag{50a}$$

$$\rho^{IE} = \frac{1}{N_E^2}\sum_{i,j\in E}^{N_E}\frac{\overline{\delta h_i^{IE}(t)\delta h_j^{IE}(t)}}{\sqrt{\Delta_i^{IE}\Delta_j^{IE}}} \qquad \rho^{II} = \frac{1}{N_I^2}\sum_{i,j\in I}^{N_I}\frac{\overline{\delta h_i^{II}(t)\delta h_j^{II}(t)}}{\sqrt{\Delta_i^{II}\Delta_j^{II}}} \;; \tag{50b}$$

where $\delta h_i^\beta(t) = h_i^\beta(t) - \overline{h_i^\beta}$ with $\beta \in (EE, EI, IE, II)$ and the overline denotes a temporal average. Analogously for the total input currents

$$\rho_T^E = \frac{1}{N_E^2}\sum_{i,j\in E}^{N_E}\frac{\overline{\delta\mu_i^E(t)\delta\mu_j^E(t)}}{\sqrt{\Delta_i^E\Delta_j^E}}, \qquad \rho_T^I = \frac{1}{N_I^2}\sum_{i,j\in I}^{N_I}\frac{\overline{\delta\mu_i^I(t)\delta\mu_j^I(t)}}{\sqrt{\Delta_i^I\Delta_j^I}}; \tag{51}$$

where $\delta\mu_i^\beta(t) = \mu_i^\beta(t) - \overline{\mu_i^\beta}$ with $\beta \in (E, I)$.

To complete the characterization of the correlations in the network we have defined also the population averaged Pearson coefficients of the firing activity of the excitatory and inhibitory neurons as

$$R^E = \frac{1}{N_E^2}\sum_{i,j\in E}^{N_E}\frac{\overline{\delta\phi[x_i^E(t)]\delta\phi[x_j^E(t)]}}{\sqrt{\Delta_i^{\phi^E}\Delta_j^{\phi^E}}}, \qquad R^I = \frac{1}{N_I^2}\sum_{i,j\in I}^{N_I}\frac{\overline{\delta\phi[x_i^I(t)]\delta\phi[x_j^I(t)]}}{\sqrt{\Delta_i^{\phi^I}\Delta_j^{\phi^I}}}; \tag{52}$$

where $\delta\phi[x_i^I(t)] = \phi[x_i^I(t)] - \overline{\phi[x_i^I]}$ and $\Delta_i^{\phi^{E,I}}$ is the variance of the firing activity of the excitatory/inhibitory neuron $i$.

## Statistics of the heterogeneous fixed point

For finite networks, depending on the network realization, the homogeneous fixed point can lose stability at $J_c$ giving rise to a heterogeneous fixed point characterized by a stationary distribution of the firing rates (input currents) $\{\phi(x_i^E)\}$, $\{\phi(x_i^I)\}$ ($\{\mu_i^E\},\{\mu_i^I\}$) and of the STD variables $\{w_i\}$. The population distribution of the input currents for sufficiently large $N$ can be assumed to be Gaussian. Therefore the stationary distribution for the excitatory (inhibitory) input currents can be obtained self-consistently by knowing the corresponding mean $\mu^E$ ($\mu^I$) and variance $\Delta_0^E$ ($\Delta_0^I$).

These can be written as follows

$$\mu^E = J_0 j_E \sqrt{N}(\sqrt{c_E}[\phi^E w] - g_E \sqrt{c_I}[\phi^I]) + I_0 \tag{53a}$$

$$\mu^I = J_0 j_I \sqrt{N}\left(\sqrt{c_E}[\phi^E] - g_I \sqrt{c_I}[\phi^I]\right) + I_0 \tag{53b}$$

and

$$\Delta_0^E = J_0^2 j_E^2 \left\{[(\phi^E w)^2] - [\phi^E w]^2) + g_E^2([(\phi^I)^2] - [\phi^I]^2)\right\} \tag{54a}$$

$$\Delta_0^I = J_0^2 j_I^2 \left\{[(\phi^E)^2] - [\phi^E]^2 + g_I^2([(\phi^I)^2] - [\phi^I]^2)\right\}. \tag{54b}$$

All the averaged values $[\cdot]$ are calculated as integrals over the Gaussian distributions, namely

$$[\phi^{E,I}] = \int \mathcal{D}z \; \phi\left(\mu^{E,I} + \sqrt{\Delta_0^{E,I}}z\right) \tag{55a}$$

$$\left[(\phi^{E,I})^2\right] = \int \mathcal{D}z \; \left(\phi\left(\mu^{E,I} + \sqrt{\Delta_0^{E,I}}z\right)\right)^2 \tag{55b}$$

$$[\phi^E w] = \int \mathcal{D}z \; \frac{\phi\left(\mu^E + \sqrt{\Delta_0^E}z\right)}{1 + \tau_D u \phi\left(\mu^E + \sqrt{\Delta_0^E}z\right)} \tag{55c}$$

$$\left[(\phi^E w)^2\right] = \int \mathcal{D}z \; \left(\frac{\phi\left(\mu^E + \sqrt{\Delta_0^E}z\right)}{1 + \tau_D u \phi\left(\mu^E + \sqrt{\Delta_0^E}z\right)}\right)^2. \tag{55d}$$

In the equations above $z$ is a Gaussian variable with zero mean and unitary variance. Notice that Eqs (55c) and (55d) are formally equivalent to Eqs (43b) and (44b) when $\tau = 0$, but unlike the rate chaos case, we can use of the fact that at equilibrium:

$$w(x) = \left(1 + \tau_D u \phi^E(x)\right)^{-1}. \tag{56}$$

The solution to this set of equations leads to the following results: below $J_c$ a unique solution with zero-variance is found which corresponds to the homogeneous fixed point reported in Eqs (5). Above $J_c$, two solutions co-exist: one corresponding to an unstable homogeneous fixed point and another one with non-zero (population) variance which corresponds to the heterogeneous fixed point.

Once the mean and variances of the inputs are found it is straightforward to derive the distribution of $w(x)$ knowing that $x \sim \mathcal{N}(\mu^E, \Delta_0^E)$. In particular the theoretical distribution takes the form

$$p_W(w) = p_X(x(w))\left|\frac{dx(w)}{dw}\right| \tag{57}$$

with $x(w)$ being the inverse function of Eq (56)

$$x(w) = \sqrt{2}\,\mathrm{erf}^{-1}\left(\frac{2(1-w)}{w\tau_D u} - 1\right)$$

with support $w \in \left(\frac{1}{1+u\tau_D}, 1\right)$. After some straightforward algebra the distribution of $p_W(w)$ is then obtained

$$p_W(w) = \frac{1}{\sqrt{\Delta_0^E \tau_D u w^2}} \exp\left(-\frac{(x(w) - \mu^E)^2}{2\Delta_0^E} + \frac{x(w)^2}{2}\right). \tag{58}$$

As shown in the inset of Fig 8C, the above distribution is in good agreement with network simulations. Using similar arguments, we can obtain the closed form of the distributions of the firing rates as:

$$p_\Phi(\phi^{E,I}) = \frac{1}{\sqrt{\Delta_0^{E,I}}} \exp\left(-\frac{(x(\phi^{E,I}) - \mu^{E,I})^2}{2\Delta_0^{E,I}} + \frac{x(\phi^{E,I})^2}{2}\right). \tag{59}$$

with

$$x(\phi^{E,I}) = \sqrt{2} \, \mathrm{erf}^{-1}\left(2\phi^{E,I} - 1\right)$$

## Lyapunov analysis

In order to characterize the different dynamical regimes emerging for finite systems at the critical point $J_0 = J_c$, we calculate the two largest Lyapunov Exponents (LE) $\Lambda_{1,2}$ associated to the model. In particular, the exponent $\Lambda_k$ has been computed by following the dynamics of the corresponding infinitesimal vector $\delta_k(t) = (\delta_k x_i^E, \delta_k x_i^I, \delta_k w_i)$ in the tangent space of the system (1). Specifically, the evolution of $\delta_k(t)$ can be obtained by integrating the system (1) together with its linearization:

$$\delta_k \dot{x}_i^E = -\delta_k x_i^E + \sum_{j \in E}^{N_E} J_{ij}^{EE}(\phi'[x_j^E] w_j^E \delta_k x_j^E + \phi[x_j^E] \delta_k w_j^E) + \sum_{j \in I}^{N_I} J_{ij}^{EI} \phi'[x_j^I] \delta_k x_j^I \tag{60a}$$

$$\delta_k \dot{x}_i^I = -\delta_k x_i^I + \sum_{j \in E}^{N_E} J_{ij}^{IE} \phi'[x_j^E] \delta_k x_j^E + \sum_{j \in I}^{N_I} J_{ij}^{II} \phi'[x_j^I] \delta_k x_j^I \tag{60b}$$

$$\delta_k \dot{w}_i^E = -\frac{\delta_k w_i^E}{\tau_D} - u(\phi[x_i^E] \delta_k w_i^E + \phi'[x_i^E] w_i^E \delta_k x_i^E) \quad \text{with} \quad k = 1, 2 \tag{60c}$$

The LEs $\{\Lambda_k\}$ can then be calculated as

$$\Lambda_k = \lim_{t \to \infty} \frac{1}{t} \log \frac{|\delta_k(t)|}{|\delta_k(0)|}, \tag{61}$$

subject to the condition $\delta_k(0) \cdot \delta_m(0) = \hat{\delta}_{km}$, with $\hat{\delta}_{km}$ representing the Kronecker delta. In practice, due to the fact that all the tangent vectors tend to align towards the largest growing direction, i.e in the direction parallel to $\delta_1$, in order to obtain the second LE $\Lambda_2$ we perform the Grand-Schimdt orthonormalization of the tangent vectors every $t_{ort} = 100$ time units by following the usual procedure introduced in [51].

## Spiking neural network: Leaky integrate-and-fire neurons with synaptic dynamics

To test the robustness of our results beyond rate models, we considered a spiking neural network composed of $N$ Leaky Integrate-and-Fire (LIF) neurons, divided into an excitatory and an inhibitory population. The connectivity matrix $J_{ij}$ follows the same block structure as in the rate model (see Eq (2)). Each neuron receives a fixed number of $K_E$ excitatory and $K_I$ inhibitory inputs from randomly selected presynaptic neurons, with no quenched disorder.

The subthreshold membrane potential $v_i(t)$ of neuron $i$ evolves according to:

$$\tau_m \frac{dv_i^{\alpha}}{dt} = -v_i^{\alpha}(t) + I_0 + \tau_m J_0 E_i^{\alpha}(t), \tag{62}$$

where the index $\alpha = \{E, I\}$ distinguishes the neuronal population. $\tau_m$ is the membrane time constant and $E_i^{\alpha}(t)$ is the total synaptic input rate stimulating the neuron $i$ within population $\alpha$. A neuron fires a spike whenever $v_i^{\alpha}(t)$ reaches a threshold value $v_{th} = 1$. After a spike, the membrane potential is reset to $v_r = 0$.

Synaptic input rates $E_i^{E,I}$ to each neuron are modeled as the linear super-position of post-synaptic potentials exponentially decaying with a time constant $\tau_{syn}$ and therefore evolve according to the following equation:

$$\tau_{syn} \frac{dE_i^E}{dt} = -E_i^E(t) + \sum_{j,f} J_{ij}^{EE} w_j(t) \delta(t - t_{f,j}^E) - \sum_{j,f} J_{ij}^{EI} \delta(t - t_{f,j}^I) \tag{63a}$$

$$\tau_{syn} \frac{dE_i^I}{dt} = -E_i^I(t) + \sum_{j,f} J_{ij}^{IE} \delta(t - t_{f,j}^E) - \sum_{j,f} J_{ij}^{II} \delta(t - t_{f,j}^I) \tag{63b}$$

where $t_{f,j}^{\alpha}$ denotes the $f$-th spike time of the presynaptic neuron $j$ of the population $\alpha$.

In the particular case of excitatory to excitatory connections, the synaptic weight is modulated by the STD term $w_j(t)$ which evolves according to following equation

$$\frac{dw_i}{dt} = \frac{1 - w_i(t)}{\tau_D} - u \sum_j \delta(t - t_{f,i}^E) w_i(t). \tag{64}$$

Each time a pre-synaptic neuron $i$ fires a spike its depression variable is updated to the value $w_i(t) \leftarrow w_i(t)(1 - u)$. Spiking simulations were performed using a Euler integration scheme with a time step $\Delta t = 1 \times 10^{-3}$, using a membrane time constant $\tau_m = 1$, and synaptic and depression time constants $\tau_{syn} = \tau_D = 10$.

## Supporting information

**S1 Appendix. Robustness of the results to the choice of transfer function.** We show that different choices of transfer functions lead to the the same asymptotic values.
(PDF)

**S2 Appendix. Finite-size corrections to the asymptotic balanced solutions.** We derive leading-order corrections to the asymptotic solutions by expanding the stationary population-averaged quantities in powers of $\epsilon = 1/\sqrt{N}$, revealing how the external current $I_0$ and synaptic coupling $J_0$ influence the dynamics in large but finite networks.
(PDF)

**S3 Appendix. Outlier eigenvalues for the homogeneous fixed point.** We illustrate how the outlier eigenvalues depend on the system size $N$, comparing the predictions of the random matrix approximation with the eigenvalues obtained from direct diagonalization of the original Jacobian matrix $\mathbf{DF}_{het}$.
(PDF)

**S4 Appendix. Validity of the DMF approximation in finite size networks.** We analyze the mismatch between the auto-correlation function (ACF) of the total inhibitory input predicted by Dynamical Mean-Field (DMF) theory and that obtained from direct network simulations. Our focus is on the role of finite network size and finite in-degree, considering sparsely

connected networks with fixed numbers of excitatory ($K_E$) and inhibitory ($K_I$) pre-synaptic connections per neuron, while systematically varying the total network size $N$.
(PDF)

**S5 Appendix. Dependence of the width of the transition region on the network size.** We analyze how the width of the transition region depends on the network size using the largest Lyapunov Exponent.
(PDF)

**S6 Appendix. Generalizations of synaptic plasticity rules: Compatibility with the self-sustained balance.** The analytical results demonstrating the existence of a non-trivial self-sustained balanced state (Eq (6)) rely on a specific architecture where Short-Term Depression (STD) acts only on excitatory-to-excitatory synapses ($J_{EE}$). Here, we explore two generalizations of the synaptic plasticity rules.
(PDF)

## Acknowledgments

The authors acknowledge useful discussions with Manuel Beiran, Nicolas Brunel, Moritz Helias, Nina La Miciotta, Antonio Politi.

## Author contributions

**Conceptualization:** David Angulo-Garcia, Alessandro Torcini.

**Formal analysis:** David Angulo-Garcia, Alessandro Torcini.

**Funding acquisition:** Alessandro Torcini.

**Investigation:** David Angulo-Garcia, Alessandro Torcini.

**Methodology:** David Angulo-Garcia, Alessandro Torcini.

**Software:** David Angulo-Garcia.

**Validation:** Alessandro Torcini.

**Visualization:** David Angulo-Garcia.

**Writing – original draft:** David Angulo-Garcia, Alessandro Torcini.

**Writing – review & editing:** David Angulo-Garcia, Alessandro Torcini.

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
