## [Decision Letter · Decision Letter 0]

20 Nov 2025

PCOMPBIOL-D-25-01707

A theory for self-sustained balanced states in absence of strong external currents

PLOS Computational Biology

Dear Dr. Angulo-Garcia,

Thank you for submitting your manuscript to PLOS Computational Biology. After careful consideration, we feel that it has merit but does not fully meet PLOS Computational Biology's publication criteria as it currently stands. Therefore, we invite you to submit a revised version of the manuscript that addresses the points raised during the review process.

We look forward to receiving your revised manuscript.

Kind regards,

Arvind Kumar, Ph.D.

Academic Editor

PLOS Computational Biology

Hugues Berry

Section Editor

PLOS Computational Biology

**Additional Editor Comments:**

The two reviewers have requested a major revision of the manuscript to clarify several points that are crucial to both results and interpretation of the results. Their comments are attached and we look forward to reading the revised version.

**Journal Requirements:**

At this stage, the following Authors/Authors require contributions: David Angulo-Garcia, and Alessandro Torcini. Please ensure that the full contributions of each author are acknowledged in the "Add/Edit/Remove Authors" section of our submission form.

4) We notice that your supplementary Figures, and information are included in the manuscript file. Please remove them and upload them with the file type 'Supporting Information'. Please ensure that each Supporting Information file has a legend listed in the manuscript after the references list.

**Reviewers' comments:**

Reviewer's Responses to Questions

**Comments to the Authors:**

Reviewer #1: In this manuscript, the authors consider networks of excitatory and inhibitory neurons in a balanced state at large system size. Unlike previous work in which external inputs to the network is strong in the limit of large network size, their model produces self-sustained non-vanishing activity in the large network limit with weak or absent external input. This is achieved with short term synaptic depression on synapses between excitatory neurons. The authors use asymptotic analysis and computer simulations to explore steady-state and non-steady state dynamics, for example chaos, in the networks. The manuscript makes a novel contribution to the mathematical analysis of these types of networks.

Major comments:

1) A central motivation for the models studied in this manuscript is that traditional balanced networks require that mean external input to each neuron scales like O(sqrt(K)) where K is the in-degree of a typical neuron. The authors are correct to point out that this has been challenged by recent theoretical work (refs 9 and 10) and that it might contradict some experimental evidence (refs 11-15), but their discussion of these points lacks nuance in such a way that it could mislead readers who are not familiar with the literature on balanced networks.

First, the authors fail to acknowledge that strong external input follows from modeling feedforward synaptic connections in the same way that they model recurrent connections: If neurons receive O(sqrt K) feedforward synaptic inputs and feedforward synaptic weights scale like 1/sqrt(K), like they have assumed of recurrent connectivity, then O(sqrt K) external input follows. If feedforward input is O(1) then feedforward synaptic inputs must either be sparser (O(1) in number instead of O(K)) or weaker (O(1/K) in strength instead of O(1/sqrt K)) than recurrent synaptic inputs. The authors should acknowledge this point and cite any evidence they have for one or the other possibility (sparser or weaker feedforward input).

On a similar note, the authors' model predicts that the average recurrent input from excitatory neurons is large (and similarly for inhibitory neurons), so feedforward input is vanishingly weak compared to each of these two sources of recurrent input. Specifically, their model assumes that recurrent excitation and recurrent inhibition are each O(sqrt K). While this fact does follow from their equations, it is never mentioned, or at least it is not mentioned where the relative strengths of feedforward and recurrent input is discussed (eg, in the second paragraph of the Introduction). The prediction that feedforward input is vanishingly small compared to recurrent input is a fundamental prediction of their model, and it should be mentioned explicitly (for example, in the second paragraph of the Introduction, after eq 5, and/or in the Discussion section).

2) In Eq 6, it is shown that steady state rates in the limit do not depend on the stimulus, I0. This would, of course, be an undesirable property of a neural network since the network should respond to stimuli. The authors later show in Fig 2 that the network does respond to the stimulus whenever N is finite and smaller in scale than one trillion (10^12). As it turns out, the asymptotic steady states in Eq 6 are actually quite inaccurate for reasonable parameter regimes (when N=10^12, neurons receive literally billions of synaptic inputs, which around a million times larger than reality). I think this point should be acknowledged. Specifically, the N=10^12 results are perhaps interesting from a mathematical perspective to verify the asymptotic equations, but the realistic values of N are much smaller, and do not adhere to the asymptotic steady state.

3) On a similar note to the previous comment, I0 does not appear in Eq 6 because it is omitted from the sqrt K scaling in Eq 5. However, a more accurate approximation could be obtained by moving the the terms I0/(sqrt(N)*J0*jX) into the parenthesis in Eq 5 and then solving the resulting linear equation to produce a version of Eq 6 that depends on N (the equation for w0 might need to be derived differently). This approximation could be plotted in Fig 2 panels C and D alongside the existing approximations. It would be worth checking whether this gives a more accurate approximation than Eq 5, particularly at larger values of I0.

4) On a related note, how does the magnitude of I0 compare to the magnitude of the total input in Fig 2 C and D? The authors' motivation for their model is that I0 is comparable to the total input (and therefore much smaller than the recurrent excitatory and recurrent inhibitory inputs individually). This is clearly accurate when I0=0 and when N=10^12, but how accurate is it for more reasonable values of N when I0 is larger? The authors could plot this ratio (either I0 versus total, or I0 versus recurrent exc) for the simulations in Fig 2C and D, either as a supplementary figure or within the main text.

Minor comments:

1) In the abstract, when the authors write "this novel balancing mechanism", it reads as if they are referring to rate chaos, whereas I am sure that they mean to refer to synaptic depression. It could be rephrase for clarity.

2) The authors consider EE short term plasticity, but not EI, IE, or II. Can the authors comment on this? Would other forms of short term plasticity achieve similar results? What if they were all combined into one model?

3) The Discussion section appears after the Methods section. This seems unusual and interrupts the reading process.

Reviewer #2: The article examines the dynamics that arise from incorporating short-term synaptic depression (STD) between excitatory–excitatory (E–E) neuron pairs in a neural rate-based network. The authors analyze the homogeneous and heterogeneous fixed points, as well as the perturbations around them. They also study the transition to chaos in finite-size networks and the convergence of this transition in the limit of large (effectively infinite) network size.

Overall, the manuscript is mathematically sound, and the derivations are clear and consistent. However, the writing and logical flow are often difficult to follow.

My main concern is that it is difficult to identify the contribution of this study. The finite-size effects discussed here are well known for rate models and can arise even without including STD. The mean-field treatment and the use of eigenvalue spectra are also standard tools in this literature. As written, it is not evident what new insight is provided specifically by the inclusion of STD. The term “massively connected” corresponds to all-to-all connectivity, although it is well established that sparse random connectivity with the same weight statistics yields identical dynamics (up to scaling by sparsity). Most of (and possibly all) the reported phenomena appear qualitatively reproducible with simpler rate models lacking STD.

For revision, I suggest that the authors clearly delineate the novel contributions of the work. In particular, they should emphasize what genuinely new dynamical phenomena arise in the large-N limit due to STD, and why these cannot be obtained from standard rate models.

**Have the authors made all data and (if applicable) computational code underlying the findings in their manuscript fully available?**

Reviewer #1: Yes

Reviewer #2: None

PLOS authors have the option to publish the peer review history of their article (what does this mean?). If published, this will include your full peer review and any attached files.

Reviewer #1: No

Reviewer #2: No

**Figure resubmission:**
---

## [Decision Letter · Decision Letter 1]

22 Jan 2026

PCOMPBIOL-D-25-01707R1

A theory for self-sustained balanced states in absence of strong external currents

PLOS Computational Biology

Dear Dr. Angulo-Garcia,

Thank you for submitting your manuscript to PLOS Computational Biology. After careful consideration, we feel that it has merit but does not fully meet PLOS Computational Biology's publication criteria as it currently stands. Therefore, we invite you to submit a revised version of the manuscript that addresses the points raised during the review process.

We look forward to receiving your revised manuscript.

Kind regards,

Arvind Kumar, Ph.D.

Academic Editor

PLOS Computational Biology

Hugues Berry

Section Editor

PLOS Computational Biology

**Additional Editor Comments:**

I apologize for the delay in the processing of your manuscript. The reviewers thought that their main concerns were adequately addressed. However, one of the reviewers commented that “I still find the manuscript difficult to follow”. This was already a concern in the first round (Q1 Reviewer #2). Therefore, I contacted the reviewer to discuss the manuscript. In addition, I also went through the manuscript more carefully. I think there are some places where the manuscript needs work both in the presentation of the results and claims that have been made.

The main claim of the manuscript is the emergence of self-sustained activity, which by definition would mean I_0 = 0. Then authors argue that a balanced state emerges for weaker feedforward inputs, unlike more classical work on this topic. While the mathematical analysis is correct, authors should note that the neuron transfer function (eq4) allows the neuron to have a nonzero response for zero input. So the claims about self-sustained activity and balanced state for weak inputs are a direct consequence of the neuron's transfer function choice. If neurons are intrinsically active (eq4) then self-sustained activity can arise even without connectivity. If the authors want to make the two aforementioned claims they must show the results for a realistic neuron transfer function (zero firing rate for zero input and strictly positive firing rate). And also please show the self-sustained activity for LIF neuron network.

Next, the results are not exclusively dependent on EEstd but different choice \phi for excitatory neurons in eq5 can also give non zero solutions.

I was also surprised by the omission of some important (maybe old) papers on self-sustained activity in balanced networks. One of the better analyses of self-sustained activity in rate based models was done by Latham et al. (J. Neurophysiol. 83(2):808-827 (2000). They showed that self-sustained activity requires intrinsically active neurons in a network. Later, Vogels and Abbott (J. Neurosci. 2005) showed self-sustained activity in a network where neurons were connected with both conductance and current based synapses. Then we (Kumar et al. 2008 Neural Computation) showed self-sustained activity in a sparse network with conductance based synapses.

The reviewer also raised an issue with the presentation of the paper. The reviewer thought that first you should show the results with N large enough to provide the "infinite " solution and then push the N-dependent solutions at the end. The focus on finite size effects is interesting but not unique to the model. I would again request you to go through the comments of the second reviewer to adjust the presentation of the manuscript.

**Journal Requirements:**

**Reviewers' comments:**

Reviewer's Responses to Questions

**Comments to the Authors:**

Reviewer #1: The authors have addressed all of my comments.

Reviewer #2: I was convinced of the interest

I still find the manuscript difficult to follow

**Have the authors made all data and (if applicable) computational code underlying the findings in their manuscript fully available?**

Reviewer #1: **No:** I can only find figures, no code or data. But maybe I am missing it somehow.

Reviewer #2: Yes

PLOS authors have the option to publish the peer review history of their article (what does this mean?). If published, this will include your full peer review and any attached files.

Reviewer #1: **Yes:** Robert Rosenbaum

Reviewer #2: No

**Figure resubmission:**
---

## [Editor Report · Decision Letter 2]

3 Feb 2026

Dear Angulo-Garcia,

We are pleased to inform you that your manuscript 'A theory for self-sustained balanced states in absence of strong external currents' has been provisionally accepted for publication in PLOS Computational Biology.

Best regards,

Arvind Kumar, Ph.D.

Academic Editor

PLOS Computational Biology

Hugues Berry

Section Editor

PLOS Computational Biology

I am happy with the revision. Both the reorganisation of the text and additional results address the concerns raised by the reviewers and the editor. Congratulations for a very fine paper.

---

## [Editor Report · Acceptance letter]

PCOMPBIOL-D-25-01707R2

A theory for self-sustained balanced states in absence of strong external currents

Dear Dr Angulo-Garcia,

I am pleased to inform you that your manuscript has been formally accepted for publication in PLOS Computational Biology. Your manuscript is now with our production department and you will be notified of the publication date in due course.

With kind regards,

Anita Estes
